# Redefining the Limits of Nanodevices-Based Drug Delivery Systems: Extracellular Vesicles

**DOI:** 10.3390/pharmaceutics17121617

**Published:** 2025-12-16

**Authors:** Marina Lucia Díaz, Victoria Simón, Luciano Alejandro Benedini, Paula Verónica Messina

**Affiliations:** 1Departmento de Biología, Bioquímica y Farmacia, Universidad Nacional del Sur (UNS), Bahía Blanca 80000, Argentina; mldiaz@criba.edu.ar (M.L.D.); mvsimon@criba.edu.ar (V.S.); 2Centro de Recursos Naturales Renovables de la Zona Semiárida, Universidad Nacional del Sur (UNS)-Consejo Nacional de Investigaciones Científicas y Técnicas, Bahía Blanca 80000, Argentina; 3Instituto de investigaciones Bioquímicas de Bahía Blanca, Universidad Nacional del Sur (UNS)-CONICET, Bahía Blanca 80000, Argentina; 4Instituto de Química del Sur, Universidad Nacional del Sur (UNS)-CONICET, Bahía Blanca 80000, Argentina; pmessina@uns.edu.ar; 5Departmento de Química, Universidad Nacional del Sur (UNS), Bahía Blanca 80000, Argentina

**Keywords:** extracellular vesicles, exosomes, engineered vesicles, nanocarriers, drug targeted delivery, RNA delivery

## Abstract

Extracellular vesicles (EVs) are naturally occurring cell-derived vesicles that contain the same nucleic acids, proteins, and lipids as their source cells. These nano-sized systems, which are derived from a wide range of cell types within an organism and are present in all body fluids. EVs play a crucial role both in health and disease, particularly in cancer and neurodegenerative disorders. Due to their particular structure, they can function as natural carriers for therapeutic agents and drugs, akin to synthetic liposomes. EVs exhibit numerous advantages over conventional synthetic nanocarriers and other lipid-based delivery systems, including their favorable biocompatibility, natural blood–brain barrier penetration, and capacity for gene delivery. However, EVs’ complex characterization and standardization, as well as being more expensive than other vesicular systems, are major drawbacks that need to be addressed before drug loading. The present review introduces the classification of EVs and their physiological roles, currently popular methods for isolating and purifying EVs, the main therapeutic approaches of EV-mediated drug delivery, and the functionalization of EVs as carriers. Consequently, it establishes novel pathways for advancing EV-based therapeutic methodologies across diverse medical disciplines. The study concludes with a discussion of the new challenges and future perspectives related to the clinical application of EVs.

## 1. Introduction

The development of pharmaceutical formulations has enabled the delivery of active pharmaceutical agents to target sites with high specificity while ensuring their protection and stability during storage. These formulations offer significant advantages, including ease of application or ingestion without the need for additional manipulation. The evolution of drug delivery systems (DDSs) has transitioned from rudimentary formulations, such as capsules or tablets, to more advanced formulations, including liposomes, niosomes, polymeric nanoparticles, and others [1]. The nanometric size of these systems has shown substantial advantages over other formulations, improving the therapeutic profiles of drugs. However, these formulations have demonstrated limitations that influence the clinical efficacy of active pharmaceutical ingredients (APIs), including suboptimal bioavailability, non-specific biodistribution, systemic toxicity, and the incapacity to traverse substantial biological barriers, such as the blood-brain barrier (BBB), among others [2]. Synthetic nanocarriers, most notably liposomes and polymeric nanoparticles, have demonstrated considerable progress in enhancing drug pharmacokinetics, API protection, targeting, and safety [3]. However, these carriers frequently encounter challenges related to immunogenicity, inefficient cellular uptake, rapid clearance by the mononuclear phagocyte system, and an absence of particular tissue targeting [4]. Moreover, a significant limitation of conventional systems is their inability to differentiate between healthy and diseased tissues, a problem that is associated with systemic toxicity and suboptimal drug levels [5]. In this context, extracellular vesicles (EVs) have emerged as an innovative and promising alternative within the field of drug delivery systems.

EVs are a diverse group of double-layered lipid particles actively produced by virtually all cell types and released into the extracellular environment. As a universal biological process, common to eukaryotes and prokaryotes, and occurs in both healthy and diseased states, thus participating in both homeostatic and pathophysiological processes [6,7]. Once thought to be insignificant cellular detritus, EVs are now recognized as crucial bioactive carriers [8]. They function as delivery vehicles for a wide array of cellular components, including proteins, lipids, and nucleic acids such as DNA, mRNA, and miRNA, facilitating complex intercellular communication and participating in numerous biological processes [9]. By transferring this cargo between cells, EVs can mediate both physiological functions or contribute to pathologies such as cancer, heart disease, and neurodegenerative disorders. EVs are a broad category that encompasses several subtypes, such as exosomes, microvesicles, and apoptotic bodies, which are classified based on their distinct biogenesis and release pathways [6]. The expanding interest in EVs has led the scientific community to develop isolation and purification techniques, as well as classification methods, based on their characteristic features [10].

After the morphology description, the first studies have shown EVs as cellular trash bags [11]. However, the posterior description of their functions has led to their consideration as potential solutions or treatments for various diseases [7], as well as therapeutic targets or natural systems for the transport of APIs [12,13,14]; due to degradation, poor pharmacokinetic profiles, the need for precise dosing, and the lack of specificity for targeting organs or tissues, a pharmaceutical formulation is required for their administration. In this context, vesicular systems have played a crucial role in advancing our understanding of biological systems and in developing safer methods for drug delivery [15].

The increasing utilization of EV-based drug delivery strategies can be attributed to their ability to circumvent the drawbacks of the conventional synthetic vesicular drug delivery systems, such as liposomes. Among these vesicles, EXs are the most common isolated for managing diseases because they can be loaded with different origin molecules. Hydrophilic (siRNAs) [16] and hydrophobic drugs as paclitaxel [17], cover and integrate into the lipid membrane, respectively. These processes protect the active agents from degradation active agents from the environment and their incorporation, depending on their physicochemical and degradative behavior, is carried out by different engineered approaches to optimize their intended properties [18]. Active and passive approaches are used for these aims. Besides, EXs have gained a significant reputation due to their biocompatibility, minimal immunogenic reactions, and extraordinary stability compared to synthetic vesicle drug delivery systems [19,20]. A selective tissue targeting mediated by surface proteins and their natural ability to cross biological barriers improves the delivery processes and reduces the off-target effects. Specific surface ligands and antibodies enhance the control and the safety of their target interaction [21]. For these reasons, EXs have been explored in a wide range of therapeutic domains, including EXs for regenerative medicine applications and engineered EXs as therapeutic devices [22,23]. One of the pioneering works about EVs as drug delivery systems has been reported by Pascucci et al. [14], in which bone marrow mesenchymal cells were exposed by in vitro assay to a high concentration of paclitaxel and then the antitumor capability of paclitaxel delivered by EXs was tested on the human pancreatic cell line CFPAC-1. Additionally, the authors demonstrated that the treated mesenchymal cells produce EXs identically to non-treated cells, keeping the EXs’ morphology. This study has shown an innovative procedure for the development of new drug delivery systems. In this scenario, a paradigm shift is emerging towards harnessing nature’s own sophisticated delivery vehicles: Extracellular Vesicles (EVs). Thus, cell-mediated delivery systems offer an alternative by leveraging the inherent properties of biological cells. These systems leverage the biocompatibility of the materials, the targeting ability, and the long circulation half-lives of cells [24].

The present review begins with EVs’ biogenesis and their main components, particularly focusing on the genetic material and proteins with relevance to EVs’ functions. We then discuss EVs’ classification and their roles in health and disease, particularly in cancer. A comprehensive discourse is presented on contemporary and extensively utilized methodologies for the isolation and purification of EVs. The principal therapeutic strategies employing EV-mediated drug delivery are analyzed, and furthermore, novel prospects for the advancement of the development of EV-based regenerative and therapeutic devices across diverse medical areas are explored. The text provides comprehensive descriptions of both native and engineered EVs, incorporating detailed explanations of the primary methodologies employed for the loading of active molecules. Finally, we address the clinical translation, the extent of progress in this regard, crucial challenges for their therapeutic application, and thus, the prospect of seeing these systems as pharmaceutical products.

## 2. From Biological Debris to Essential Cell Language

A fundamental revolution in our understanding of cell biology is based on the discovery of intracellular communication. The shift in the concept of EVs from junk to essential cellular machinery is primarily driven by new insights into their critical functions. For decades, what we now call EVs have been seen as cellular trash bags or debris because they participate in cellular housekeeping, mainly committed to the constant turnover of cell components such as proteins, lipids, and organelles. For this reason, it was thought that EVs were merely an ejection system of unwanted material out of the cell. Another type of debris, apoptotic bodies, which are formed by parts of cells after programmed cell death, should fulfill another cleaning-up step by immune cells like macrophages. The milestone that changed the EV paradigm was described in the late 90s, when EXs isolated from immune cells were found to activate the immune system by maturing antigen-presenting cells (APCs) and T lymphocytes. Reinforcing this new change of paradigm, the transfer of different RNAs, including miRNAs, among cells was the most relevant fact. In this context, it was demonstrated that this transfer of genetic material led to the expression of new proteins in different species of mammalian cells [7,25]. This sequence of events, followed by recent findings, has led to a shift in the perception of EVs, which are now considered essential structures with crucial roles in both health and disease.

## 3. Advances in Understanding the Biology and Biogenesis of Extracellular Vesicles

The formation and release of EVs is a complex and highly regulated process involving numerous genes and proteins that determine their identity, cargo, and function and generally fall into three interconnected categories [26]. These components include structural regulators of membrane remodeling [27,28,29], adaptors for nucleic acid sorting [30,31] and trafficking mediators [32,33]. Together, they enable the transformation of EVs from passive cellular debris into dynamic nanodevices capable of intercellular communication [34], cargo delivery [25,35], and disease modulation [36]. Understanding this machinery is essential to harnessing EVs for therapeutic applications [37] and to decode their role in pathological contexts such as cancer [36], neurodegeneration [38], and immune dysregulation [39].

### 3.1. Exosomes

The main subtypes studied for therapeutic purposes are exosomes (EXs) because of their size, low immunogenicity, and targeting capabilities. These vesicles show a single lipid bilayer membrane from 50 to 150 nm formed through internal vesicles within the endocytic system, where the endosomes mature into late endosomes and form multivesicular bodies (MVBs), whose intraluminal vesicles fuse with the plasma membrane for release [40,41] (Figure 1). EXs are found in biological fluids such as serum, platelet-rich plasma [42], urine [43], cerebrospinal fluid [44], breast milk [45], and saliva [46]. They are produced by cell cultures of different types, from dendritic and macrophages to mesenchymal cells. Each EX shows properties associated with yield production and particular morphological and structural features of each family of EXs. Among EX-producing tissues/organs, it is important to distinguish between healthy and ill tissues, such as cancer cells, since in the latter, the overexpression of particular surface biomarkers is commonly observed [47].

One of the most well-characterized mechanisms driving intraluminal vesicle formation within multivesicular bodies involves the endosomal sorting complex required for transport (ESCRT). The ESCRT comprises four core modules—ESCRT-0, ESCRT-I, ESCRT-II, and ESCRT-III—along with the AAA ATPase Vps4 (vacuolar protein sorting–associated protein 4) and multiple accessory regulators. Together, these components coordinate the recognition of ubiquitinated cargo, its concentration within phosphatidylinositol 3-phosphate (PI3P)– and lipid-enriched endosomal microdomains, membrane deformation, and the scission events that release intraluminal vesicles (ILVs) into the multivesicular body (MVB) lumen [29,48,49]. ESCRT-0 initiates cargo clustering at PI3P-rich patches via subunits containing ubiquitin-interacting motifs (UIMs) and FYVE domains; ESCRT-I reinforces this clustering and recruits ESCRT-II, which in turn scaffolds membrane curvature and nucleates ESCRT-III assembly through interactions between VPS25 (vacuolar protein sorting–associated protein 25) and CHMP6 (charged multivesicular body protein 6). ESCRT-III polymers—primarily CHMP4, with regulatory input from CHMP2, CHMP3, CHMP1, and CHMP5—constrict the neck of nascent ILVs, and Vps4 disassembles these filaments to complete membrane fission and recycle components [27,50,51,52]. The CHMP proteins are cytosolic ESCRT-III subunits that polymerize upon activation to drive membrane remodeling. Several accessory proteins modulate the spatial and temporal dynamics of this machinery. HD-PTP (His domain–containing protein tyrosine phosphatase) acts as a multivalent scaffold that bridges early and late ESCRT components, directing ubiquitinated receptors such as EGFR toward lysosomal degradation. Its loss or dysfunction alters receptor fate and promotes aberrant signaling [53,54,55]. Ist1 (increased sodium tolerance 1 homolog), another ESCRT-III–related protein, regulates filament dynamics and Vps4 activity, influencing whether membranes undergo scission or are recycled, and thereby tuning the balance between degradative and recycling pathways [52]. Lipid composition also plays a decisive role: cholesterol- and sphingolipid-rich microdomains, along with lysobisphosphatidic acid (LBPA), define membrane regions permissive for ILV budding and influence adaptor recruitment, linking membrane chemistry to pathway selection.

In parallel to the canonical ESCRT-0/I/II cascade, an alternative route involving ALIX (ALG-2–interacting protein X) and syntenin can directly engage ESCRT-III. Although this pathway bypasses the upstream ESCRT modules, it remains mechanistically dependent on ESCRT-III polymerization for intraluminal vesicle formation. ALIX, via its Bro1 domain, binds CHMP4 to promote filament assembly, while syntenin (syndecan-binding protein with PDZ domains) interacts with syndecans to couple specific cargo, such as tetraspanins, to budding sites. This route is responsive to signaling cues and lipid composition, particularly LBPA enrichment, and contributes to the functional diversification of exosome content [56,57]. Importantly, although this ALIX–syntenin axis bypasses upstream ESCRT components, it should not be confused with ESCRT-independent mechanisms, as it still relies on ESCRT-III activity. Together, these canonical and alternative ESCRT-dependent pathways form a flexible network in which cargo identity, adaptor availability, and membrane context converge to determine the site and mechanism of ILV formation. ESCRT-III polymerization executes the mechanical constriction, while regulators such as Ist1 and HD-PTP fine-tune the outcome toward degradation, recycling, or secretion. This coordinated machinery not only governs the physical formation of exosomes but also shapes their molecular composition and downstream biological roles. In addition to these ESCRT-dependent mechanisms, exosome biogenesis can also occur through ESCRT-independent routes. These include ceramide-induced membrane remodeling via nSMase2 [58], tetraspanin-enriched microdomains involving CD63, CD81, and CD9 that organize membrane topology and cargo selection [59,60], and the regulatory role of bioactive lipids such as phosphatidylcholine, phosphatidic acid (PA), and ceramide in modulating membrane curvature and vesicle release [61]. These alternative routes contribute to the diversity of exosomal content and are discussed in greater detail in Section 3.5.

### 3.2. Microvesicles (Ectosomes)

Microvesicles (MVs) or ectosomes (ECs) are plasma membrane-derived systems with a range of sizes from 100 to 1000 nm, and they arise from an external protrusion of membrane (Figure 1). Therefore, they reflect the composition of the plasma membrane more closely than EXs. Particular biomarkers, such as integrins, selectins, CD40, and tissue factor, are shown by ECs. While almost any cell can produce ectosomes as much as exosomes, the main EC producers are highly active cells, undergoing stress, or involved in dynamic communication processes. The main sources of healthy cell-derived ECs include platelets, due to their abundance in the blood, immune cells like macrophages, and endothelial cells. Among diseased cells, cancer cells are the most notable. The function of the ectosome is entirely dependent on the state and identity of the parent cell it comes from. ECs’ biogenesis begins at the plasma membrane through a highly coordinated process in which lipid reorganization, membrane organizers, and small GTPase-driven cytoskeletal remodeling act as interdependent modules that coordinate membrane protrusion, neck formation, and eventual scission [62,63,64]. Local enrichment of cholesterol- and sphingolipid-rich microdomains concentrates cargo and lowers the energetic barrier for membrane curvature, while phosphatidylserine externalization destabilizes the bilayer and promotes budding [65,66]. Tetraspanins, such as CD9, CD63, CD81, and CD82, assemble into microdomains that organize membrane topology and recruit specific cargo and linker proteins, for example, CD82 recruiting ezrin, thereby coupling cargo clustering to sites of membrane deformation [41,67]. Small GTPases integrate membrane cues with actin dynamics: RhoA–ROCK (Ras homolog family member A–Rho-associated coiled-coil containing protein kinase) signaling increases actomyosin contractility to drive membrane blebbing and large vesicle release; Cdc42 (cell division control protein 42 homolog) coordinates localized actin polymerization via effectors like IQGAP1 (IQ motif containing GTPase activating protein 1) to shape protrusions and directional shedding; and ARF6 (ADP-ribosylation factor 6) triggers a PLD → ERK → MLCK cascade (phospholipase D, extracellular signal-regulated kinase, and myosin light chain kinase) that enhances cortical contractility for scission and modulates cargo selection [66,68,69]. Mechanistically, these modules interact dynamically: tetraspanin- and lipid-defined microdomains bias where GTPases and their effectors act; GTPase activity remodels lipid organization and cortical tension; and cytoskeletal forces concentrate and assist in severing tetraspanin- and lipid-rich buds, producing microvesicles whose size, cargo, and release kinetics reflect their combined lipid composition, scaffold assembly, and GTPase signaling.

### 3.3. Other Extracellular Vesicles

Apoptotic Bodies (ABs) are another type of EVs that also come from the plasma membrane, produced by cytoplasmic fragmentation during programmed cell death (apoptosis) (Figure 1). They contain cellular debris and are generally less interesting for targeted drug delivery due to their random content and their role in waste disposal. ABs are larger than those mentioned and range from 100 to 5000 nm. ABs exhibit biomarkers as annexin V, C3b, thrombospondin, Annexin A1, and histone coagulation factor [70].

Novel groups of EVs include exomeres, which are EVs measuring ≤50 nm secreted by cleavage of large cytoplasmic extensions of cells, migrasomes measuring 500 to 3000 nm that form at the tip or at the bifurcation of retraction fibers during cell migration, and have Tspan4, CD63, and Annexin A1 as biomarkers. Large Oncosomes measuring 1000 to 10,000 nm are released by cancer cells with amoeboid movement. Supermers measuring approximately 35 nm (<50 nm) with a still unknown origin and mechanism.

There are also engineered vesicles formed by the combination of exosomal membranes with synthetic nanoparticles, called hybrid vesicles. These systems combine the advantages of both systems to improve stability, carrying capacity, and targeting precision. Furthermore, EVs can also be classified according to the physiological state of the cells of origin, such as “oncosomes” released from cancer cells or “prostasomes” originating in the prostate [70,71]. These systems (hydrides and oncosomes) will not be addressed in this text.

### 3.4. Molecular Cargo of Extracellular Vesicles

EVs, including both EXs and MVs, carry diverse biomolecules that reflect the physiological or pathological state of their cell of origin. Their cargo comprises membrane and cytosolic proteins (tetraspanins, heat shock proteins, enzymes, receptors), lipids (ceramide, cholesterol, phosphatidylserine), metabolites and amino acids, and nucleic acids, including genomic and mitochondrial DNA and a range of RNA species: messenger RNAs (mRNAs), microRNAs (miRNAs), long non-coding RNAs (lncRNAs), circular RNAs (circRNAs), small nuclear RNAs (snRNAs). This complexity enables EVs to act as vehicles of intercellular communication that can modulate gene expression, immune responses and signaling in recipient cells. In this section, we focus on the selective sorting and packaging of nucleic acids into EVs, emphasizing experimental evidence and methodological caveats.

Although EVs in general can carry RNA and DNA, preferential accumulation of particular RNA species has been most robustly documented for exosomes; thus, mechanistic extrapolation to microvesicles should be made cautiously because direct, causal evidence for many sorting pathways in microvesicles remains limited [72]. The content is summarized in Table 1.

#### 3.4.1. RNA Sorting and Packaging in Extracellular Vesicles

The selective RNA loading into EVs, particularly exosomes, is orchestrated by a network of RNA-binding proteins (RBPs) that recognize specific sequence or structural motifs and couple RNA cargo to the vesicle biogenesis machinery. Experimental evidence from perturbation studies, including knockdown, overexpression, post-translational modification (PTM) mutants, and affinity capture, has demonstrated that these RBPs influence the enrichment of defined RNA subsets in EVs, although the causality and universality of each mechanism remain context-dependent. Among the best-characterized RBPs is heterogeneous nuclear ribonucleoprotein A2B1 (hnRNPA2B1), which binds miRNAs containing GGAG motifs and requires SUMOylation for its sorting activity [30]. Additional PTMs, such as O-GlcNAcylation of its RNA recognition motif, modulate its RNA-binding affinity under stress [73]. hnRNPA2B1–RNA complexes have been observed to associate with ceramide-rich microdomains and multivesicular body (MVB) membranes, although these interactions appear model-dependent [83,84]. Another key player, Y-box binding protein 1 (YBX1), facilitates the sorting of diverse RNA species, including miRNAs, tRNA fragments, Y RNAs, and long mRNA fragments, by forming phase-separated condensates that concentrate RNA cargo and direct it to MVBs [31,74]. YBX1 activity is modulated by PTMs and cellular context, such as stress or signaling cues, which influence its sorting efficiency [85]. Argonaute 2 (AGO2), a core component of the RNA-induced silencing complex (RISC), has also been detected in EVs bound to miRNAs; its perturbation reduces the export of specific miRNAs, and oncogenic KRAS–MEK–ERK signaling alters AGO2 phosphorylation and subcellular localization, thereby modulating its incorporation into small EVs [75,76]. However, AGO2 is not universally required for miRNA loading across all systems [31]. More recently, heterogeneous nuclear ribonucleoprotein K (hnRNPK) has emerged as a multifunctional adaptor that binds purine-rich RNA motifs and assembles into ribonucleoprotein complexes targeted to endosomal membranes. Its localization to enlarged MVBs and interaction with membrane-remodeling proteins such as caveolin-1 suggest a direct role in exosome biogenesis [57,77]. PTMs further regulate hnRNPK activity, influencing the selective packaging of miRNAs and small nucleolar RNAs (snoRNAs) into intraluminal vesicles (ILVs). Functional perturbation of hnRNPK alters the abundance of its RNA targets in EVs and impacts downstream signaling in recipient cells, reinforcing its role in cargo selection [35]. Collectively, these RBPs operate within a dynamic and interconnected system, where their modifications, interactions with membrane domains, and responsiveness to cellular signals converge to shape the RNA landscape of secreted vesicles.

Regarding microvesicles (ectosomes), emerging evidence suggests that they also exhibit selective RNA cargo profiles, although through distinct and less characterized pathways. As previously described, their biogenesis at the plasma membrane involves lipid remodeling and cytoskeletal dynamics that generate specialized microdomains enriched in tetraspanins and signaling lipids, which may serve as platforms for RNA recruitment [41,67]. RNA-binding proteins such as hnRNPA2B1 and YBX1 have been detected in microvesicle fractions, and their association with lipid rafts and actin-linked scaffolds suggests alternative sorting routes independent of endosomal machinery [31,78,79]. Additionally, small GTPases such as ARF6 and CDC42, which regulate membrane protrusion and scission, may indirectly influence RNA cargo by modulating the localization of sorting adaptors and the timing of vesicle release [68,86]. Although direct causal links between specific RNA-binding proteins and RNA content in microvesicles remain to be fully elucidated, the interplay between membrane topology, lipid composition, and cytoskeletal remodeling appears to bias the inclusion of defined RNA subsets, contributing to the functional heterogeneity of EV populations [66].

#### 3.4.2. DNA Sorting and Packaging in Extracellular Vesicles

The selective incorporation of DNA into extracellular vesicles (EVs) involves a set of mechanisms that reflect both passive capture and active recruitment of genetic material from the nucleus and mitochondria. DNA cargo can appear as double-stranded, single-stranded, or fragments of chromatin, and its presence has been documented across the different EV types including exosomes, microvesicles, and apoptotic bodies [80,87]. While some DNA is surface-associated or co-isolated with non-vesicular nucleoprotein complexes, a fraction is genuinely intravesicular, as demonstrated by nuclease protection and density-gradient fractionation assays [72,88]. Mechanistically, chromosomal fragments released during micronucleus formation or nuclear envelope rupture may enter the cytosol and be packaged into vesicles via endosomal or autophagy-linked routes [6] DNA-binding proteins such as histones and high-mobility group box proteins (HMGB1/2) stabilize DNA and facilitate its tethering to membrane domains or sorting complexes, contributing to both structural packaging and immunostimulatory potential in recipient cells [87]. In regard to the mitochondrial DNA (mtDNA), it is selectively incorporated through interactions with mitochondrial nucleoid proteins, including transcription factor A (TFAM), which protect and guide mtDNA into exosome-enriched fractions [81]. ESCRT-associated adaptors such as TSG101 and ALIX, though classically linked to protein sorting, may also participate in the recruitment of DNA–protein complexes under conditions of cytosolic DNA accumulation [80]. Additionally, markers of DNA damage and micronucleus dynamics, such as γH2AX and other nuclear envelope rupture indicators, correlate with elevated EV-DNA release, suggesting a stress-responsive packaging route. Candidate nucleases and DNA-binding chaperones are still being explored and it is likely that they influence the size, integrity, and vesicular localization of DNA cargo, although their specific roles remain under investigation [72]. Functionally, EV-associated DNA can activate cytosolic sensors such as cGAS–STING in recipient cells, triggering innate immune responses [82], and in some tumor models, EV-associated DNA has been shown to transfer oncogenic sequences, though durable horizontal gene transfer in vivo remains controversial [80]. Altogether, these findings suggest that DNA packaging into EVs is governed by a context-dependent interplay between nuclear integrity, stress signaling, membrane remodeling, and DNA–protein interactions.

### 3.5. Secretion of Extracellular Vesicles

The final step in the extracellular vesicle (EV) lifecycle, their secretion into the extracellular space, is governed by a specialized set of molecular regulators that orchestrate vesicle trafficking, docking, and membrane fusion. This process is not merely a passive release but a tightly controlled event that determines the spatial and temporal availability of EVs, their interaction with recipient cells, and their functional impact in physiological and pathological contexts. Key players include Rab GTPases, which act as molecular switches to direct vesicular compartments toward the plasma membrane [32,89], and tetraspanins, which organize membrane microdomains and influence cargo sorting and vesicle uptake [90,91]. Lipid-modifying enzymes such as neutral sphingomyelinase 2 (nSMase2) and phospholipase D2 (PLD2) further modulate membrane curvature and lipid composition, facilitating vesicle release through ceramide- and phosphatidic acid–driven pathways [58,61]. These secretion mechanisms are responsive to cellular stress, signaling cues, and disease states, underscoring their relevance in EV-mediated communication and their potential as therapeutic targets in cancer, neurodegeneration, and immune disorders [92,93,94].

#### 3.5.1. Rab GTPases

The small Rab GTPases represent a family of roughly 70 proteins within the Ras GTPase superfamily. They act as molecular switches by cycling between GTP- and GDP-bound states. In their active, GTP-bound conformation, Rab proteins interact with effectors to regulate multiple steps of vesicular transport and direct membrane trafficking. Rab GTPases are widely regarded as organelle-specific markers, as each Rab protein governs a particular step in intracellular trafficking [89]. The most important Rab subfamily members involved in EV secretion are Rab11, Rab35 and Rab27. Rab11 and Rab35 coordinate the recycling of membrane components from endosomes back to the plasma membrane, a process that can facilitate several cellular processes such as cytokinesis, migration, and neurite extension [95,96,97]. In contrast, the secretory Rab27 is dedicated to directing late endosomal and lysosome-related compartments toward the plasma membrane [32].

The Rab11 subfamily consists of three members: Rab11a, Rab11b and Rab11c (also known as Rab25). Rab11a, the first characterized, is ubiquitously expressed [98], whereas Rab11b and Rab25 display more restricted tissue distribution [99,100]. While the Rab11 subfamily plays a key role in slow recycling membrane components, Rab35 controls a rapid endocytic recycling pathway that returns multiple proteins, such as transferrin, to the plasma membrane [101]. Rab27 is broadly conserved across metazoans, with vertebrates expressing two isoforms, Rab27A and Rab27B, that participate in the regulated secretory pathway. Unlike Rab27, exosomal secretion regulated by Rab11 and Rab35 appears to be ESCRT-independent, supporting the idea that this pathway serves as a mechanism for disposing specific cellular components; on the other hand, Rab27-regulated exosomal secretion might play a role in intercellular communication [101]. Whether these GTPases are constitutively regulating exosomal secretion or are recruited under certain (pathological) conditions is still a major question to be clarified.

#### 3.5.2. Tetraspanins: CD9, CD63 and CD81

Tetraspanins are a conserved protein family present in all metazoans, comprising 33 members in mammals. They are relatively small (200–300 amino acids), variably glycosylated proteins characterized by four transmembrane domains [102,103]. Tetraspanins display diverse subcellular localizations: while proteins like CD9, CD81, and CD151 are mainly found at the plasma membrane [91,104,105], others such as CD63 and Tspan6 reside intracellularly, trafficking through late endosomes and lysosome-related organelles [106,107,108]. They are also enriched in specialized curved or tubular structures, including microvilli, tunneling nanotubes, migrasomes, retraction fibers, and virus budding sites [60,67].

Tetraspanins are abundantly present in EVs, where CD9, CD81, and CD63 serve as widely recognized markers. While CD63 is often linked with exosomes [59], CD9 and CD81—typically located at the plasma membrane—are more commonly associated with microvesicles [91,109], though all three can be detected in both EV subtypes as well as in other extracellular structures, including midbody remnants and migrasomes [67,110]. Their role in EV formation, cargo sorting, and functional properties (uptake, fusion, cell migration) has been studied extensively, but results remain contradictory, suggesting that tetraspanins are not essential for EV biogenesis and may act in a context-dependent manner [90]. Nonetheless, specific tetraspanins can influence protein sorting into EVs (e.g., CD 82 and CD9 with β-catenin [111]; CD9 with metalloproteinase CD10 [112]; Tspan8 with E-cadherin [113]. In fact, several studies suggest that CD81 may play a role in both protein and miRNA cargo selection in EVs [30,114]. Moreover, tetraspanins might also be involved in the organization of lipids such as sterols within these vesicles [90]. Tetraspanins play a crucial role in regulating physiological processes that include signal transduction, cellular activation, motility, adhesion, and tissue differentiation. These processes are controlled directly by the regulation of cellular interactions (cell membrane-bound molecules) or indirectly through exosomes [115].

#### 3.5.3. Lipid-Related Genes

In 2008, it was reported that EVs can also form through an alternative ceramide-driven pathway [58]. A key physical feature of ceramide is its strong intrinsic negative curvature, derived from its small polar head group, together with its ability to cluster into ceramide-rich microdomains that promote a shift in membrane organization from a planar lamellar state towards a more compact hexagonal phase [116]. Local accumulation of ceramide at specific membrane regions facilitates the inward budding that gives rise to ILVs [58]. Ceramide is primarily generated through the hydrolysis of sphingomyelin (SM), which yields ceramide and phosphorylcholine, a reaction carried out by members of the sphingomyelinase (SMase) family. Sphingomyelinases comprise a heterogeneous group of enzymes that catalyze SM hydrolysis, and they are classified mainly according to their optimal pH, cation dependency, and subcellular distribution. Among them, four nSMases (types 1 to 4) have been described, with nSMase2 being the most extensively characterized. This isoform is abundantly expressed at the plasma membrane and Golgi in various mammalian cells, with particularly high levels in the brain [117,118]. Although transient elevations of nSMase2 activity are part of normal physiology, sustained activation of this enzyme and the consequent rise in EV release have been associated with several pathological contexts, including modulation of immune responses during brain inflammation [94], cancer metastasis [92], amyloid accumulation [119,120], spreading of Tau protein [93], and HIV infection [121]. In line with these findings, both genetic silencing and pharmacological inhibition of nSMase2 have been shown to suppress EV secretion [93,94]. Collectively, this evidence indicates that targeting nSMase2 to limit EV release could represent a promising therapeutic strategy in diseases where EV-mediated intercellular communication plays a detrimental role.

Reports have also described a third, ESCRT- and ceramide-independent mechanism of ILV budding, which relies on the small GTPase ADP ribosylation factor 6 (ARF6) and its downstream effector, phospholipase D2 (PLD2) [61]. Phospholipases D (PLD) comprise a family of 6 members in mammals, with isoforms PLD1 and PLD2 being the most studied and characterized. These enzymes catalyze the hydrolysis of phosphatidylcholine, producing phosphatidic acid (PA) and choline. PA, the simplest phospholipid, plays a crucial role in membrane dynamics such as fission and fusion [122]. Its activity is largely attributed to its small headgroup, which induces negative membrane curvature and forms a cone-like shape that favors intraluminal budding within endosomes. Moreover, lipid-binding studies have shown that syntenin directly associates with PA [61]. Beyond curvature effects, this interaction may help recruit syntenin/syndecan/CD63/ALIX complexes to nascent buds, where ALIX–ESCRT machinery further enhances the budding process. PA may also support ILV budding independently of ESCRT. Notably, nSMase2 has been reported to bind PA [123,124], an interaction thought to facilitate its targeting to MVBs and stimulate local ceramide production, which further contributes to EVs [124]. In addition, PA may contribute to a positive feedback loop with SRC: by binding to SRC, PA promotes its open, active conformation [125]. Activated SRC can then phosphorylate and activate PLD2 [126], reinforcing PA production and thereby sustaining ILV biogenesis, in agreement with SRC’s recently established role in exosome formation [127]. Notably, both ceramide and PA promote EV formation through their capacity to induce negative membrane curvature and reorganize local lipid domains, thereby facilitating inward budding [58,61]. In addition, each lipid can serve as a signaling hub, recruiting specific protein complexes—ceramide through SMase activity and PA through syntenin and SRC–PLD2 feedback—that reinforce ILV biogenesis [61]. These parallels suggest that lipid-driven mechanisms converge on common biophysical principles while engaging distinct molecular partners to sustain EV production.

Categorization of RNA and DNA cargo by subtype, cellular origin, and sorting pathways, highlighting key RNA-binding and DNA-associated proteins involved in vesicle biogenesis and cargo selection. Functional implications refer to the biological roles of these nucleic acids in recipient cells, such as gene regulation, immune activation, and disease modulation. RBPs: RNA-Binding Proteins; PTMs: Post-Translational Modifications; MVBs: Multivesicular Bodies; dsDNA: Double-Stranded DNA; ssDNA: Single-Stranded DNA; mtDNA: Mitochondrial DNA; TFAM: Transcription Factor A, Mitochondrial; HMGB1/2: High-Mobility Group Box Proteins 1 and 2; AGO2: Argonaute 2; cGAS–STING: Cyclic GMP–AMP Synthase–Stimulator of Interferon Genes pathway; γH2AX: Phosphorylated H2A Histone Family Member X (marker of DNA damage).

## 4. Extracellular Vesicles and Their Role in Cancer

Whether released from healthy or diseased tissues, EVs can carry pathogenic substances that may contribute to the onset or progression of various disorders. In the context of cancer, tumor cells have been shown to secrete larger quantities of EVs -mainly exosomes- enriched with diverse molecular cargo, compared with normal cells [87]. Exosomes function as intercellular messengers that coordinate nearly every aspect of cancer biology, from initiation and angiogenesis to immune escape, metastasis, and therapy resistance. Their presence in body fluids also makes them valuable as non-invasive biomarkers and potential therapeutic targets [128]. There are several major mechanisms by which EVs promote cancer progression. In the tumor initiation process and proliferation, cancer stem cells (CSCs) represent a small fraction of tumor cells; however, their self-renewal and differentiation abilities are frequently associated with tumor relapse, as their capacity for dormancy allows them to later reawaken and restart tumor growth. CSC-derived EVs promote tumorigenesis, metastasis, and therapy resistance by transferring malignant traits from donor cells [129,130]. In the ever-growing evidence from different cancer types, a pivotal role for miRNAs containing cancer EVs is emerging, particularly in colorectal [131], lung [132], skin [133] and breast [134], among many other cancers. Tumor-derived EVs can stimulate endothelial cell migration, proliferation, and tube formation while disrupting endothelial junctions to increase vascular permeability. This is known as angiogenesis and vascular remodeling in the oncologic process. Studies from gastric [135], hepatocellular carcinoma [136] and other human cancer cells [137] have demonstrated that tumor derived- EVs induced VEGF secretion, thus promoting angiogenesis. Moreover, exosomes from hepatoma cells can affect the integrity of endothelial junctions and promote vascular leakage [138]. A specific process related to metastases is the Epithelial-to-Mesenchymal Transition (EMT). The EMT is a conserved biological process in which epithelial cells lose their characteristic features and acquire mesenchymal traits, leading to alterations in cell shape and behavior. This transition represents a crucial early event in tumor cell invasion and the subsequent development of distant metastases. Tumor-derived EVs loaded with TGFb, miR-21 or metalloproteinases—among many other cargoes- induce EMT to enhance invasiveness, chemoresistance and metastatic potential [139,140].

EXs generate Immune evasion by influencing a wide range of immune cells, including cytotoxic T cells, NK cells, macrophages, neutrophils, myeloid-derived suppressor cells (MDSCs), dendritic cells (DCs), and regulatory T cells. Their immunomodulatory activity is primarily driven by the presence of immune-associated molecules, such as non-coding RNAs, proteins, and surface components like Major Histocompatibility Complex class I and II, or T cell-activating factors [141]. Exosomes contribute to tumor progression by promoting an immunosuppressive environment. They can impair T cell responses and NK cell cytotoxicity while enhancing the activity of myeloid-derived suppressor cells MDSCs. In addition, tumor-derived EXs may hinder T cell proliferation and function either directly or through the suppression of DCs activity [141]. These EVs condition distant organs to favor metastatic colonization by modifying stromal, immune, and endothelial cells [142], which is considered a pre-metastatic niche formation. A pioneering work from Hoshino and colleagues demonstrated that the integrin profile of tumor-derived EVs displays organ-specific tropism, showing a preference for certain target organs over others [143]. This selective pattern suggests that integrin signatures on EVs could serve as valuable biomarkers for predicting organ-specific metastasis.

The tumor microenvironment (TME) comprises cellular and molecular components—such as stromal cells, extracellular matrix and signaling factors—that interact dynamically with cancer cells to regulate tumor growth, metastasis, and prognosis [144,145]. The tumor microenvironment remodeling is a crucial process for the advancement of cancers. In hepatocellular carcinoma (HCC), this immune microenvironment plays a crucial role, as impaired immune surveillance, and host immune suppression drive disease aggressiveness. Evidence indicates that exosome-mediated signaling stimulates an immunosuppressive niche, enabling tumor cells to evade immune destruction and thus promote cancer progression [146,147,148]. Cancer-derived EVs can alter glucose and lipid metabolism in recipient cells, favoring tumor growth and survival, which leads to a metabolic reprogramming of cells. EVs released from differentiating cancer stem cells can activate cancer-associated fibroblasts and facilitate tumor progression by stimulating aerobic glycolysis [149]. EVs can also enhance cancer cell migration through fatty acid oxidation [150]. Prostate cancer–derived EVs promote metabolic reprogramming in recipient cells, enhancing glycolysis and survival under acidic conditions through activation of key glycolytic enzymes. This mechanism supports tumor cell proliferation and resistance within the tumor microenvironment [151]. Among the cell reprogramming, tumor-derived EVs can promote drug resistance by spreading drug-resistance genes, miRNAs, and proteins [152]. Their cargo can also induce the export of anti-tumoral drugs out of tumor cells; EVs can transport resistance-associated molecules—such as ATP-binding cassette (ABC) transporters- that facilitate drug efflux [153,154]. Cancer EVs can also transport anti-apoptotic proteins and regulatory miRNAs that suppress apoptosis while promoting DNA repair [152].

Detecting and characterizing tumor-derived-EVs circulating in the bloodstream offers great promise for their use as biomarkers in cancer disease diagnosis and prognosis. As research on EVs expands, their potential as both therapeutic tools and diagnostic agents continues to grow. Their natural role in transferring molecular information between cells highlights their importance and versatility in biomedical applications.

## 5. Pharmaceutical Implications of Biological Features of EVs

The molecular components described in the above sections are not merely markers or biological features of EVs; they are functional determinants that directly shape their pharmaceutical potential in drug delivery systems (DDSs). These important structures are mainly proteins: Tetraspanins, ESCRT Complex, Rab GTPases, and ribonucleoproteins. In this section, we focused on structures that can modify the properties of EVs for pharmaceutical applications.

The first group is tetraspanins (CD9, CD63, CD81). They are biomarkers that facilitate vesicle formation and provide high efficiency of cellular uptake, providing the natural tissue tropism, and can be engineered to enhance cell-specific delivery of EVs, playing a crucial role in targeting. Their presence ensures reproducibility as a quality control process, fits EVs into a specific classification, and meets particular properties of EVs to be defined as active targeting DDSs [60,155]. Second, the ESCRT Complex, which is the machinery that drives inward budding of endosomes to form multivesicular bodies (MVBs), leading to exosome release. This protein complex naturally determines the cargo selection (being proteins and RNAs) to be packaged into EVs. Consequently, the manipulation of ESCRT can enhance and enable the loading of particular active agents, such as small molecules or different types of RNAs [156]. Another group of proteins with a significant biological role regulating the vesicle trafficking, docking, and secretion of EVs, and with relevance in drug delivery systems, are Rab GTPases such as Rab27, Rab11, and more. These proteins control the release efficiency of EVs from parental cells, a crucial step for “production”; for that reason, engineering Rab pathways can increase EV yield, improving scalability for pharmaceutical production. The increase in yield is a key factor to industrial-scale EV manufacturing and consistent therapeutic dosing [157]. Finally, ribonucleoproteins such as hnRNPA2B1 that participate in stabilization and sorting of RNAs into EVs gain their relevance in delivery systems because they ensure selective packaging of therapeutic RNAs (siRNA, miRNA, mRNA), protecting them from degradation. This fact increases the stability and circulation time and directly affects gene therapy applications of EVs [158].

## 6. Isolation and Purification Methods

This is a foundational technical section, as the choice of isolation method directly impacts the yield, purity, and functionality of the obtained EVs, which is a major challenge for clinical translation. Isolation and purification methods are based on the consideration of the main structural characteristics of EVs: size (density), charge, and affinity [159].

The purification is strictly associated with the aim of the posterior application of EVs. For example, for diagnosis, highly purified product is not necessary, but high-yield isolation is. EVs from ectosome or endosomal origin share similar size and densities, and the process became more complicated due to plasma lipoproteins also sharing these features [41]. Two of the most widely used isolation approaches are ultracentrifugation and precipitation-based techniques. The first one has been the “gold standard,” involving high-speed centrifugation to pellet EVs based on their size and density under centrifugal force The procedure requires a considerable amount of time (3–4 h); however, it exhibits low scalability and high costs [160]. The precipitation-based technique, is a simple and high-yield procedure, based on differential solubility, which shows low costs and high scalability. However, along with EVs, they often co-precipitate “contaminants” such as lipoproteins and consequently, require additional careful washing [161]. Additionally, it requires a high sample processing time and high sample volumes. The techniques analyzed in this work are described separately, but a combination of two or three of them has been reported for additional purity and separation. For example, size-exclusion chromatography (SEC) and centrifugal ultrafiltration have been combined for isolating EVs from culture media and other EVs of different origins [162].

In this section, we will address size-exclusion chromatography (SEC), filtration (centrifugal and tangential flow ultrafiltration), affinity-based techniques, ion-exchange techniques, electrophoresis, and microfluidics.

### 6.1. Size-Exclusion Chromatography (SEC)

Size exclusion chromatography (SEC) is a proven technique primarily applied to macromolecules, such as proteins, due to its ability to preserve EV integrity and functionality. This technique selects components based on their molecular size or hydrodynamic volume. According to the components of chromatography, the system involves a stationary phase, porous in this case, which can be coupled or not to a pump. A scheme of this technique is shown in Figure 2. One of the main features of SEC is its capability to solve a wide variety of samples from different origins, including eukaryotes and prokaryotes. However, the main difference between these EVs produced by these cells is based on their lipid membrane composition, membrane proteins, and cargo, but they share a range of sizes. Therefore, it is important to note that this technique can be applied to both, but it cannot distinguish between different cell origins of EVs. This technique can isolate EVs from varied samples such as milk, saliva, and synovial fluid. One of the main challenges for SEC isolation capability is when the sample exhibits complex fractionation, contains lipoproteins and proteins, and we are working with diluted samples; therefore, it should be concentrated. The necessity for standardized protocols will help with the problems based on the sample origin. The scalability is another limitation to be considered [163]. To solve some of these drawbacks, urinary sub-populations of exosomes can be separated by applying a two-step SEC. This approach, described as two-dimensional SEC, employs two consecutive SEC columns: the first, with a pore size of 50 nm, retains larger particles and thereby allows the smallest EVs to elute first, while the second column, with 200 nm pores, further resolves the remaining, larger vesicle fractions [164]. The method uses agarose beads as the stationary phase, which are easily adjustable and scalable for different sample types. Bed volume and exclusion limits are selected according to sample requirements; for example, Sepharose^®^ CL-4B with 42 nm pores is preferred over 75 nm pores when separating EVs from proteins such as albumin. Consequently, SEC offers advantages over ultracentrifugation, which is unable to efficiently isolate and separate these components [165,166]. Many columns can be chosen depending on the original material. One of the most versatile columns is the IZON^®^ qEV column, which can be used for cell culture [167], plasma or serum [168], urine [169], saliva and tear fluid [170], milk [171], lymph node and spleen [172]. Sepharose^®^ CL-2B, Sepharose^®^ CL-4B, and Sephacryl^®^ S-400 columns are also used for isolating EVs from cell cultures and from other biological sources [159].

### 6.2. Filtration

Filtration membrane technique or ultrafiltration is a size-based and widely used for the isolation and purification of extracellular vesicles. It is an advancement of conventional filtration and is suitable for diluted samples. The system uses filtration membranes housed in a container. Vesicles isolated by this method have a molecular weight ranging from 10 to 100 kDa. The technique is easy to use, simple, fast, can process large sample volumes, and does not require specialized equipment [173]. A schematic representation is shown in Figure 3. However, this technique shows several drawbacks, including long processing times, potential for protein aggregate contamination, retention of EXs onto the membrane, and shear forces that may cause damage in EVs or morphological changes [173,174].

Additionally, membrane blockage decreases its efficiency and reduces the lifetime, which confers another problem due to their high costs. Additional filtration-based approaches are also available. Tangential flow filtration relies on the same fundamental principle and reduces this drawback. In this method, the flow is parallel (tangential) to the membrane surface rather than perpendicular, as occurs in centrifugal ultrafiltration. As a result, molecules smaller than the membrane pore are discarded, and bigger such as EVs, remain on the membrane and are subsequently collected with a previous concentration by recirculation. This technique gains importance for diluted samples [175]. Sequential ultrafiltration (SU) is another similar technique that involves a three-step filtration procedure. First, debris, second, proteins, and last, EVs are separated [176]. For exosomes isolation and separation, SU employs hydrophilic polyvinylidene fluoride membranes with progressively decreasing pore sizes from 0.65 to 0.1 μm. This technique can be combined with filtration (macrofiltration, of bigger vesicles such as apoptotic bodies), but it must be considered that the limitations of including this technique as an intermediate phase [177].

### 6.3. Affinity-Based Techniques

Affinity-based techniques are one of the most commonly used for isolation and separation of EVs. They utilize highly selective and specific interactions between ligands, mainly proteins, or receptors on the surface of EV membranes and their corresponding ligands, such as antibodies [159]. To carry out this technique, the ligands must be immobilized or conjugated onto a solid medium. A scheme is shown in Figure 4. For this aim, solids support as magnetic beads, polymeric materials, or polysaccharides such as agarose, are used. These media are distributed or packed in a column. Affinity-based isolation techniques are broadly divided into two categories—affinity chromatography and immunocapture—based on their distinct isolation mechanisms. Immunoaffinity capture (or immunocapture) is the predominant affinity-based technique for EV isolation. It employs antibodies coupled covalently to magnetic beads, designed to bind to specific proteins present on the EV membrane. As previously mentioned, tetraspanins CD9, CD63, and CD81 are the main biomarkers on the surface of EXs working as antibody targets. Immunoaffinity is advantageous in facilitating the selective isolation of EVs from different healthy and cancerous cells. Several cancer cells from lungs, stomach, and hormone-dependent tissues overexpress the glycoprotein CD326, the epithelial cell adhesion molecule (EpCAM). Thus, anti-EpCAM conjugated magnetic beads were used to isolate EXs from cancerous origin cells. This strategy has been used for the isolation and separation of tumor-derived EXs from healthy cell-derived EXs of plasma [178]. There are many antibodies proposed for the selective isolation of EVs from different tissues. The anti-A33 [179], anti-CD171 [180], anti-CD105 [181], and anti-CD41 from colon epithelial, neuron-derived, endothelial cell, and platelet-derived [173], respectively. For these reasons, this technique exhibits higher specificity and yield compared with filtration techniques. However, “biased” isolation is a frequent drawback of this method. EXs expressing a particular biomarker are detained by the column, but others will pass through. It is known that surface heterogeneity (CD9+ vs. CD9−) is most likely a reflection of deep functional heterogeneity. If a study uses only anti-CD9 to isolate exosomes and then reports a function, it is reporting the function of that specific subpopulation (CD9+), which may be only part of the whole story [182].

Similar to EXs, microvesicles (MVs) or ECs show differential surface phospholipidic biomarkers, annexins V and A1, that have been used for their isolation [159]. Unlike the other mentioned techniques, once the formation of complex EV-beads is produced, two ways can be taken. One, just after the isolation, the complex can be washed and retrieve the EVs and continue with the experiments; or two, the complex can be stored for future assays, later, without compromising the integrity of the EVs [183]. The are different solid supports used in immunoaffinity chromatography, such as magnetic beads, which have been mentioned, and additionally polysaccharides, such as cellulose [184]. One of the most popular devices is the monolithic support, which has been used for biomacromolecule separation (lipoproteins) and EVs. Among its advantages, the monolithic disk with immobilized antibodies can be used many times, it shows short isolation times, and it can withstand alkaline pH conditions, and it can be coupled with other separation methods [161].

### 6.4. Ion-Exchange Techniques

These techniques, such as ionic-exchange chromatography and metal-affinity, are based on ion-exchange because the procedures imply an interaction between the negative surface charge of vesicles, which is mainly given by phospholipids, and a positively charged anion exchanger. Once bonded with the support, EVs are released by the use of high salt concentration solutions, as eluent or mobile phase, which increases the ionic strength of the media (Figure 5). The main difference between the two techniques involves the retention of the EVs. In ionic-exchange chromatography, the retention is produced by resins, and in the metal-affinity technique, by metal or magnetic beads, which are removed along with bonded EVs by a magnet. The columns used in this type of chromatography can show a strong anion exchanger, such as a monolithic column with quaternary amine functionality, or a weak anion exchanger, such as a diethylaminoethyl cellulose resin. This technique has been applied to the isolation of mesenchymal stem cell-derived EVs [185], amniotic fluid [186], and cell cultures such as HEK293T cells (kidney embryonic cells) [187]. However, the applications of Anion exchange chromatography as a single isolation technique are limited to cell cultures despite its versatility and scalability. Kosanović et al. have reported the application of this technique for the isolation of EVs combined with ultracentrifugation [186]. This technique shows several limitations when samples show a high amount of charged molecules, such as proteins. For that reason, samples of blood or derived from blood sometimes will need a complementary method to improve the purity.

### 6.5. Electrophoresis

This technique is based on the separation of EVs according to their electrophoretic mobility, which is a function of their surface charge-to-hydrodynamic radius ratio. Since electrophoresis is a measure of how fast a charged particle moves through a gel matrix under the influence of an electric field, this technique provides additional information about EVs, such as charge, which is not provided by other techniques based on density or size measurements [159]. Additionally, it is a fast technique, and high EX purity can be reached by combining it with size-based methods. Lan et al. [188] have reported the application of electrophoresis for the isolation of EXs from urine fluid. The sample was previously treated with a density gradient separation method, and then two subpopulations of EXs were isolated by electrophoresis. EXs from human plasma and HeLa cell cultures have also been isolated [189]. In other instances, electrophoresis is coupled with additional preparatory strategies to enhance EX separation. For example, El Ouahabi et al. [190] reported isolating EXs from human plasma using PEG 8000 and a low-speed centrifugation step before electrophoresis

### 6.6. Microfluidic

This is an emerging technique that allows the control and manipulation of small volumes of fluids, typically ranging from nanoliters to microliters, within microscale networks of channels excavated by a laser [191]. In this context, microfluidic devices have been used for EXs separation because they show advantages such as reduced volumes of sample (and buffer) due to the miniaturization, the option to have the device custom-built, and high throughput capability [192]. These devices show the additional advantages of microscale, such as improved thermal transport, low contamination risks, minimized time-dependent degradation reactions by reducing the separation or analysis time, and others. Therefore, this technology, allows a rapid and efficient isolation and detection of EVs from complex biological samples. This is an advantage over other techniques such as SEC. Therefore, it streamlines the process, shortening the analysis periods of a large number of samples and improving high-throughput screening [193]. Microfluidic devices were first thought with capillary systems for electrophoresis, but nowadays are based on a chip format. Separation of EXs can be boosted by the immobilization of ligands (affinity-based methods), mainly antibodies or proteins, on the channel walls or compartments of the device or by incorporating membrane filtration, such as nanoporous membranes [194] (Figure 6). Affinity-based isolation methods coupled with a microfluidic device can be automated for sample processing, avoiding expensive and complex devices. An affinity-based microfluidic device functionalized with antibodies of membrane biomarkers CD9 and EpCAM, which was used for small EXs isolation from high-grade serous ovarian cancer serum [195]. A microfluidic device based on membrane filtration was developed by Zhao et al. [196] for automatic isolation and enrichment of EXs formed by two membranes. The first one removes large-sized particles and blood cells, while the second, positively charged, retains the EXs by complementary charge. This approach reached a 99.8% protein impurity removal and isolated more than 80% of EXs from the samples. Therefore, the adaptation of membranes to these devices enables the separation of EXs from different origins such as human plasma, bronchoalveolar fluid, and urinary fluid [197].

Finally, it is important to notice that the separation methods previously described in the above sections can be incorporated into microfluidic devices for use as separation and isolation methods; however, the costs of microfluidic devices can increase depending on the complexity of the selected approach, and protracted processing time can be attributed to the small volumes of samples.

It is worth mention that further analysis is required once EVs are isolated, thus characterization techniques are imperative to ensure vesicle properties. Some of the most recognized methodologies for characterizing EVs are: Transmission Electron Microscopy (TEM) for morphology and size, Dynamic Light Scattering (DLS) and zeta potential for size (polydispersity of the sample) and surface charge, Western Blot, flow cytometry and ELISA for analyzing specific biomarkers such as CD9, CD63, CD8, PCR for nucleic acids, among other techniques [198].

## 7. Therapeutic Approaches of EV-Mediated Drug Delivery

Based on the extensive description of EVs previously presented, it is possible to infer two crucial features: first, EVs are formed by a complex combination of components that allow their free traveling all over the body; and on the other, they can deliver many different cargoes, from nucleic acids, proteins, and small molecules to tissues [6]. These two features, besides their low immunogenicity, biocompatibility, biodegradability, target specificity, and stability derived from their origin and the structural and morphological properties, lead us to think that they could become one of the most studied structures for drug delivery systems [199]. These properties are the first considered when a pharmaceutical device for drug delivery is designed, and in this case, they are naturally fulfilled by these vesicles. Besides, the innate biocompatibility and low immunogenicity are a crucial advantage with respect to synthetic membrane carriers or lipid-based carriers, for which different modifications, such as PEGylation (coating with polyethylene glycol) [200], should be done to avoid the immune system, which can trigger inflammatory responses needing treatment. However, some patients can develop anti-PEG antibodies [201]. Therefore, the rise of these eventual drawbacks enhances the consideration of EVs as DDSs.

The arrival of EVs at the target cell can be mediated by the biomarker (natural tropism), but the release of their content is warranted by merging the membrane with the target cell or by phagocytosis [202]. Therefore, harnessing the presence of surface biomarkers, different treatments could be potentially addressed by using EVs with particular surface targeting proteins. Active and passive immunological acquisition (vaccination or antibodies), gene therapy, anti-cancer therapies, treatment, diagnosis, and tissue regeneration are possible approaches [6].

Another important aspect of EV biology is the mechanism by which they release their cargo. EVs can reach target cells through specific biomolecular recognition (natural tropism), and once internalized, their content is delivered either through direct membrane fusion or via phagocytic uptake. A key functional advantage of EVs is their ability to escape the endosomal–lysosomal pathway. Because they have evolved specifically to transfer cargo between cells, EVs possess intrinsic mechanisms that allow them to evade lysosomal degradation. In contrast, synthetic vesicle-based nanocarriers generally lack this capability and often fail to escape lysosomal sequestration [203].

A notable benefit of EV-based delivery systems is their capacity to cross biological barriers, which represent significant obstacles for other delivery systems (such as synthetic). One such example is the Blood-Brain Barrier (BBB), where these vesicles have been shown to naturally cross, rendering them well-suited to the treatment of neurological disorders (Alzheimer’s, Parkinson’s, brain cancers) [204]. It is important to note that the mechanism by which EXs diffuse across the BBB remains to be fully elucidated. However, the extant literature suggests that exosome transport through the BBB occurs primarily through transcytosis and is influenced by several regulators, such as inflammation and metastasis [205].

Since ECs are formed by direct budding from the plasma membrane is harder to control and standardize their genesis to obtain a reproducible delivery system. Their larger size can also be a limitation for extravasation and penetration, but might be useful for loading larger cargo. Many therapeutic studies use a mixture of EXs and ECs because full separation is technically challenging. However, the field is moving toward using EXs for more precise and reproducible therapeutics [206]. There are different strategies for using EVs as drug delivery systems. One of them is based on the application of non-modified EVs (natural), and the other, on EVs engineered by different strategies. Both approaches will be discussed in the next section (Section 8).

## 8. Engineered EVs and Non-Engineered EVs for Therapeutic Activity, and Regenerative Medicine

### 8.1. Non-Engineered EVs

As previously mentioned, EVs are naturally loaded with different specific molecules; thus, unmodified EVs (or non-engineered EVs) can show therapeutic activity. Mesenchymal cells show the ability to differentiate, self-renew, develop immunomodulatory properties, and promote tissue reparation [207]. Therefore, mesenchymal cell-derived exosomes (MSCEx) also play a crucial role in physiological processes such as angiogenesis, metabolism, immunomodulation, and inflammation. Vonk et al. [208] have reported that bone marrow MSCEXs inactivate the tumor necrosis factor-α (TNF-α)-mediated upregulation induced by cyclooxygenase-2 (COX2) and pro-inflammatory interleukins, inhibit the collagenase activity, which decreases the collagen remodeling, and produce type II collagen and proteoglycans. These observations directly imply cartilage regeneration and the potential of being applied as therapy against osteoarthritis. The inflammatory bowel disease is characterized by a loss of intestinal barrier integrity, which is reversed by the application of bone marrow MSCEXs by increase the IL-10 and TGF-β levels and the decrease in vascular endothelial growth factor (VEGF)-A, TNF-α, the interleukin IL-12, and promotes M2 macrophages activity [209,210] Thus, as a proof of concept, EXs isolated without modifications could be applied as therapeutic devices for treating different diseases.

### 8.2. Engineered EVs

Another approach to the therapeutic application of MSCEXs is based on the development of engineered EVs. Non-engineered EVs can show therapeutic activity, but this can be potentiated because of their capability to transport synthetic substances or from different origins. The selection of the active loaded substance categorizes the EVs, and the loading strategy differs significantly for each category [211]. Additionally, EVs can often be functionalized to unlock the full potential of EVs, moving them from passive delivery vehicles to active targeting systems [212]. This is essential for treating complex diseases like cancer, where specific cell targeting is paramount. This strategy is designed to target engineered EVs to specific tissues or cells, enhancing their therapeutic efficacy and reducing off-target effects. One of the most common strategies is Parent Cell Engineering: The source cells are genetically engineered to express targeting ligands (e.g., peptides, antibody fragments, receptors) on their surface. These ligands are naturally incorporated into the membranes of the EVs they produce. Another approach is carried out when EVs have already been isolated and involves chemical or physical methods to attach targeting moieties to their surface, such as inserting ligand-anchored lipids, peptide conjugation, and others. The goal is to express molecules like RGD peptides (to target integrins in tumors), neurotropic factors (to enhance brain uptake), or VCAM-1 targeting peptides (to target inflamed endothelium) on the EVs surface; and thus, achieving a specific disease treatment, avoiding off-target effects [213].

#### 8.2.1. Loading Methods of Active Agents

Considering the nature of the loaded material and types of EVs, engineering EVs can incorporate interesting genetic material (both DNA and types of RNAs), proteins, and small molecules (drugs) by several strategies. Due to that, loading processes show different approaches. One of them is viral transfection, which is based on the use of a virus, such as a lentivirus, adenovirus, retrovirus, or other. This is the most effective method for loading siRNA, miRNA, long non-coding RNAs, and proteins [214], and it is the preferred method for bone marrow MSCEVs and DC-EVs, as it minimizes damage to the EVs. Lentivirus and adenovirus are the most commonly used for therapeutic EVs due to their efficiency and safety profiles. Lentiviruses are mainly used for the transfection of HEK293T cells, dendritic cells, and adenoviruses for neuronal and muscle cells. Additionally, the choice of virus depends on two main factors: the cell type and the desired stability of expression [215]. Each cell type has a different tolerance and efficiency against a viral vector. Consequently, MSCs tolerate adenovirus and lentivirus well, while neurons prefer adenovirus due to its low immunogenicity. Besides, this last virus does not integrate significantly, but may maintain prolonged expression in cells such as neurons [216]. It is notorious that the transfected structure is the parental cell; therefore, the genetic material is included in the cell before EVs are generated [10]. Once transfected, the genetic material is naturally packaged into EVs during their biogenesis, applying the cell machinery discussed in Section 5.

Another approach is the non-viral transfection, which is a group of techniques based on diverse strategies and shows a differential efficiency depending on the molecule to be loaded. Additionally, some of them can also be used for loading genetic material into EVs, not only for small molecules. Non-viral transfection methods are applied after the EVs are formed and purified [217]. Among the techniques applied for this aim, we mention: the loading with cationic molecules, such as cationic lipids or polymers, or those using physical processes, such as electroporation and ultrasonication. The RNAs, which are negatively charged molecules, can be transfected by forming a complex with a positively charged molecule. Lipofectamine was used to effectively load siRNA into EXs to knock down specific genes in recipient cancer cells. However, this technique requires the application of an additional purification after the transfection due to the complex formation [218,219].

Otherwise, sonication or ultrasonication is mainly applied to the load of hydrophobic molecules such as paclitaxel [220]. The loading mechanism is based on the disruption of the membranes by ultrasound waves, which can provide an energetic force that can alter the EVs’ structures. This drug has been loaded by sonication in EXs of the human pancreatic cancer cell [220], EXs of RAW 264.7 macrophages [221], and EXs derived from U-87 glioblastoma cells [17]. Finally, electroporation is a loading technique frequently used for hydrophilic small molecules such as doxorubicin or small RNAs [222]. Its mechanism is based on the application of electric pulses, increasing the EX membrane permeability, allowing the entrance of molecules. Mesenchymal stem cells [223], HEK293F (kidney) [224], and human breast cancer cell lines MDA-MB-231 [225] are the main cells loaded by this technique.

#### 8.2.2. Engineered EVs Loaded with Active Agents

Peptides and proteins are naturally loaded and delivered by EXs, and therefore, these vesicles could be loaded with proteins searching for different objectives, for example, as replacement therapies, tissue regeneration, cancer, vaccines, and more [226]. In this context, hormones, recombinant proteins, and monoclonal antibodies can be used to interact with specific targets within the body for different treatments [227]. To prevent loss of activity due to enzyme degradation or clearance, proteins must be protected from the environment. In this context, different synthetic pharmaceutical formulations, such as hydrogels and lipid-based formulations, have been developed [228]. Besides degradation, another important concept that should be considered in loading proteins is the hydrophilic-lipophilic behavior. The partition and the charge of a protein play a crucial role, while lipophilic proteins will immerse into the EV membrane, hydrophilic ones concentrate in the core of the EVs, and amphiphilic ones show a dual behavior. Besides, charged proteins can bind or generate repulsion by electrostatic interactions [229].

The Lysosome-Associated Membrane Protein 2B (LAMP2B), anchored to bone marrow mesenchymal cells (MSCEXs) resulted useful for the transportation of the ischemic myocardium targeting peptide (IMTP), which was effectively targeted to in vitro model of ischemic myocardium using H9C2 cells are an immortalized cell line derived from myoblasts (muscle precursor cells) from the right ventricle of rat embryos [230]. Once the in vitro model demonstrated the targeting, the system was applied to in vivo assays, using female C57BL/6 mice. These loaded EXs showed a reduction of inflammation with a reduced infarct zone and cardiomyocyte apoptosis, and improved angiogenesis in the infarcted zone. These results suggest that a better and faster recovery would be expected.

Tendon remodeling is another significant challenge to overcome because it is inserted in a hostile mechanical environment, exhibits low vascularization and cell density for remodeling, and displays a dense collagen matrix that is injured and replaced with scar tissue [231]. In this context, rat platelet-derived EXs have been loaded with a recombinant Yap1 protein (which regulates cell growth) to promote the rejuvenation of tendon stem cells. This protein exhibits lipophilic and hydrophilic portions, which hinder the loading [232]. Therefore, alternative processes or techniques for improving the loading of such molecules must be developed. Protein-loaded EXs have also been explored for cancer therapy, including applications in breast cancer. In an in vivo model using electroporation-loaded EVs, a triphenylphosphonium (TPP)–modified recombinant therapeutic tumor suppressor protein, P53, was encapsulated within tumor cell–derived EVs. The TPP moiety directed P53 to the mitochondria of breast cancer cells, inducing cell death [233].

Nucleic acid- EVs’ loading for gene therapy is one of the most prominent applications, as EVs are nature’s own gene delivery vehicles. Small Interfering RNAs (siRNA) and microRNAs (miRNA), used to knock down or silence the expression of specific disease-causing genes and to alter gene expression networks, in the second case, show particular application in oncology, genetic disorders, and antiviral therapies. In the regeneration of certain tissues, such as cardiac tissue or in neurodegenerative diseases, miRNAs play a particular role. The payload of plasmid DNA in EXs has also been reported, but less frequently, due to difficulties in getting into the nucleus [226]. Genetic material is negatively charged due to phosphate groups and shows hydrophilic behavior. Even containing hydrophobic portions, these portions form an interior portion where hydrophilic and charged portions are exhibited to the outside. Due to these structural properties, the use of genetic material for treatment, such as gene therapy, displays several limitations and challenges. Although different delivery systems have been implemented, the in vivo performance does not fulfill the objectives with additional toxicity states and immune responses. In this sense, EVs have demonstrated biocompatibility and the capability to naturally transport genetic material. In addition, EXs can be engineered, increasing the selectivity by anchoring or favoring the expression of particular surface molecules or biomarkers, and for this reason, reducing off-target effects [229]. In this context, certain miRNAs have been loaded into MSCs or other cells to suppress tumor growth by inhibiting oncogenic pathways. In a collagen-induced arthritis mouse model, Chen et al. [234] have injected engineered MSCEXs containing the plasmid miR-150-5p by transfecting MSCs. In this pathology, the dysregulation of miRNAs in synovial fibroblasts, osteoclasts, and T lymphocytes mediates joint destruction. By this strategy, the authors have demonstrated the reversal of increased activity of matrix metalloproteinases, fibroblast-like synoviocytes, and angiogenesis factors associated with arthritis reumathoidea.

Ohno et al. [235] have engineered exosomes from the HEK 293 line (a human embryonic kidney cell line) for delivering the miRNA let-7a, a tumor suppressor. The GE11 peptide was anchored on the surface of exosomes to bind the epidermal growth factor receptor expressed on cancer cells. This peptide can be clinically used instead of epidermal growth factor. Therefore, in this study, the authors have shown a reduction of tumor growth by application of engineered EXs with the GE11 peptide as miRNA delivery carrier in RAG2-/mice model of cancer.

The skin is the largest organ of the human body, and some generalized skin conditions can lead to severe health issues. Those suffering skin damage from ultraviolet radiation or chemotherapy need an effective response. One strategy to improve cutaneous photosensitivity and dark pigmentation after chemotherapy is based on the inhibition of genes such as NF-kB, AP-1, and MMPs, which play a crucial role in skin lesions, improving skin self-healing ability. Thus, siRNA against genes (NF-kB) in skin lesions has been loaded into EXs derived from adipose-derived mesenchymal stem cells genetically engineered by donor cell transfection with lentivirus [16]. The study has demonstrated both in vitro and in vivo results, where the activity of EXs loaded with siRNA against NF-κB was evaluated in epithelial cells and macrophages (RAW 264.7 cell line). A significant decrease in NF-κB expression was observed, as well as a reduction in macrophage proliferation. Using a skin lesion model in C57BL/6 mice exposed to UVB irradiation, the authors demonstrated that mice treated with siRNA-enriched EXs showed faster recovery from skin lesions, which contributed to the repair and remodeling of damaged skin cells. In another study, adipose-derived mesenchymal stem cells were used to produce EXs loaded with the mRNAs of human vascular endothelial growth factor A (VEGF-A) and human bone morphogenetic protein 2 (BMP-2) for enhancing both angiogenic–osteogenic regeneration. These EXs were then loaded within a customized injectable PEGylated poly (glycerol sebacate) acrylate (PEGS-A) hydrogel for bone regeneration in rats with challenging femur critical-size defects [236]. Therefore, in this work, the protein and nucleic acid were loaded into EVs as two therapeutic agents to reach a global regeneration of the tissue. In another study, Wan et al. [237] have developed engineering EVs from human embryonic kidney cells (293T) loaded with Cas9 ribonucleoprotein (RNP) for treating a type 1 Herpes Simplex Virus infection. For boosting the efficiency of the RNP, the robust interaction between Fc (human immunoglobulin fragment crystallizable) and SpA (Staphylococcal protein A) was harnessed. The authors propose the eradication of the 1 Herpes Simplex Virus by spCas9 RNP, which is a desirable effect not reached by antivirals. Finally, the efficacy of both in vitro and in vivo assays was demonstrated for the EVs.

Small molecules or active pharmaceutical ingredients (API) can be delivered by EVs, such as traditional chemotherapeutic agents, doxorubicin, paclitaxel, and methotrexate, anti-inflammatory drugs, such as dexamethasone and hormones, and others. Engineered or modified drug-loaded EVs involve loading therapeutic cargo into EVs after isolation. There are two primary EV charging approaches: active and passive. The first one involves active procedures, such as electroporation (electrical pulses to create temporary pores), sonication (sound waves), or extrusion (mechanical force) to facilitate cargo entry. Passive approaches are only applied to hydrophobic drugs because they involve diffusion across the lipid membrane when co-incubated. We will address examples of drug-loaded EVs, mentioning the loading process and their therapeutic roles.

Cancer is the second primary cause of death. Chemotherapeutic drugs show several adverse effects; therefore, their specific targeting is crucial for improving the treatments. Doxorubicin causes cardiac and bone marrow toxicity, which is dose and time treatment dependent. EXs derived from a kidney cell line (HEK293T) and engineered with peptides Angiopep-2 and TAT to target the brain. Angiopep-2 is a peptide for crossing the blood-brain barrier, and TAT is a transcription transactivator. Then, these EXs were loaded with doxorubicin via electroporation for treating gliomas. These EXs have significantly improved survival time in mouse models without serious side effects [238]. Another example of the doxorubicin chemotherapeutic capability delivered by EXs has been described by Kim et al. [239]. In this work, Signal-Regulatory Protein alpha (SIPRα)-loaded EXs derived from HEK293T were designed to block the tumor antigen CD47, which is associated with cancer immunotherapy. This antigen is overexpressed in tumor cells and blocks the phagocytic activity of immune cells. In this case, doxorubicin was loaded into these EXs targeting their anticancer effect to the surface-expressed CD47. In this case, doxorubicin was incubated directly with EXs for its loading, demonstrating in vitro activity.

Paclitaxel is another anticancer drug loaded into EXs. This taxoid shows many adverse effects, which could be minimized with specific targeting devices. These effects are produced by the paclitaxel itself, such as bone marrow suppression, but also by the solvent used for its dissolution, chremophor, which generates severe allergies. To avoid the use of this solvent and improve the targeting, paclitaxel has been loaded into EXs of a human pancreatic cancer cell line (PANC-1) by direct sonication with EXs. This system has demonstrated a significant reduction in tumors in the cancer xenograft model mice [220]. With the same aim, paclitaxel has been loaded into a hybrid formed by EXs derived from mesenchymal stem cells and liposomes through repeated freeze–thaw cycles. Loaded EXs have demonstrated enhanced treatment efficacy in colorectal tumor-bearing mice models. Additionally, they increase the activation of CD4+ and CD8+ T cells and modulate the tumor immune microenvironment [240].

Methotrexate is another drug that has severe adverse effects, such as neuronal damage, bone marrow suppression, hepatic injury, and renal dysfunction, which is widely used for anti-cancer and immunogenic therapy. High doses of this drug are used for central nervous system lymphoma treatment with a high ratio of these adverse effects. The capability of EXs to cross the blood-brain barrier loaded with methotrexate shows an attractive alternative to the application of this drug. Zhao et al. [12] have carried out experiments with EXs derived from human adipose-derived mesenchymal stem cells with an engineering surface with an anti-CD19. The CD19 antigen is highly expressed on the surface of malignant mature B lymphocytes, which has proven effective in treating various B-cell malignancies such as B-cell non-Hodgkin lymphoma. One achieving the goal of crossing the blood-brain barrier for treating glioma, EVs derived from fibroblasts have been loaded with methotrexate. Moreover, to improve the uptake by glioma cells, the EVs were functionalized with therapeutic [Lys-Leu-Ala (KLA)] and targeted [low-density lipoprotein (LDL)] peptides. The effectiveness of the proposed treatment was evidenced both in vitro by 3D glioma spheroids of U87 cells and in vivo BALB/c mouse model [241].

Using a mouse model (6 and 8-week-old BALB/c strain) of autoimmune hepatitis, Zhao et al. [13] have shown that dexamethasone, into EXs secreted from mesenchymal stem cells, has the ability to reverse the concanavalin A-induced autoimmune hepatitis (AIH). The EXs can suppress concanavalin A (Con A)-induced liver injury, and combined with dexamethasone, this effect can be improved (synergy).

Osteoporosis is caused by an imbalance between bone tissue formation and resorption, often driven by inflammatory processes. The use of estrogens is a therapeutic strategy for treating it; however, their use can cause different diseases, including cancer. Thus, EXs loaded with 17β-estradiol by incubation and sonication with an accurate targeting could be a rational option for the treatment of osteoporosis, avoiding the off-target effects. EXs derived from bone marrow mesenchymal cells were used with this aim, showing a significant increase in cell survival by the MTT method [242].

## 9. Clinical Translation of Extracellular Vesicles as Drug-Delivery Systems

The clinical use of extracellular vesicles (EVs) as drug-delivery systems has made significant progress. However, this area is influenced by both promising early clinical results and the substantial challenges of production, characterization, regulatory review, and large-scale implementation. More than thirty interventional studies are looking into mesenchymal stromal cell (MSC)-derived EVs. These studies focus on a variety of conditions, including acute inflammatory responses like graft-versus-host disease and COVID-19-related acute respiratory distress syndrome. They also cover chronic conditions such as Crohn’s disease, osteoarthritis, neurodegeneration, and tissue repair after stroke or spinal cord injury [243]. Applications in oncology are also emerging, particularly regarding the delivery of chemotherapy drugs or vesicles containing tumor antigens that can boost antitumor immunity [244].

While the range of conditions shows the biological flexibility of EVs, it highlights the need for a consistent framework for translation in this field. Dosing regimens vary widely. Intravenous infusion is the most common method. It usually involves particle numbers between 10^9^ and 10^12^ per dose, given weekly or biweekly based on the clinical context [245]. Local methods, such as intranasal, intra-articular, intrathecal, or intradermal administration, are increasingly used to enhance tissue exposure and decrease systemic clearance [246]. Overall, safety outcomes in trials have been mostly positive, with most adverse events being mild and self-limiting. However, differences in EV composition and the lack of standard dosing metrics limit how outcomes can be compared. This highlights the need for consistent potency units or activity-based dosing criteria [247]. In the early phases of clinical trials, the focus is on safety, tolerability, immune activation markers, or early functional improvement. More advanced trials will need validated surrogate markers, PK/PD relationships, and disease-specific clinical endpoints.

Safety considerations extend beyond clinical observations; controlling the manufacturing process is also essential. One of the most important standardization patterns is their heterogeneity based on physicochemical features and the capability to identify variants of EVs. The current isolation techniques do not guarantee the homogeneity of production batches, a crucial factor in the transition from investigational states to the clinic. Additionally, the yield is also affected by the approach applied in isolation. Two or more techniques are sequentially used to improve the purity but this approach negatively impacts performance [248].

Variability in donor MSCs, changes in EV composition due to culture, and the risk of including unwanted bioactive materials like oncogenic RNAs or angiogenic factors are significant concerns [246]. Therefore, workflows that follow good manufacturing practices (GMP) include careful donor screening, cell bank qualification, monitoring of manufacturing conditions, removal of apoptotic bodies and protein aggregates, sterility tests, and endotoxin limits. Engineered EVs add risks related to membrane manipulation, unintended activity of foreign materials, and uncertainty about ligand-engineered or genetically altered vesicle surfaces [249,250]. This demands specific tests for membrane integrity, aggregation, ligand stability, and dose-dependent toxicity in suitable preclinical models, as required by EMA (European Medicines Agency) and FDA (Food and Drug Administration) guidelines for advanced biological products.

Chemistry, Manufacturing, and Controls (CMC) requirements are crucial for ensuring product consistency and meeting regulatory standards. For example, potency tests need to reflect the mechanism of action. Immunomodulatory EVs require assays for macrophage polarization, cytokine-release suppression, or T-cell inhibition. In contrast, genetic-cargo EVs depend on transfection efficiency or reporter-gene expression for functional readouts [244]. Purity criteria rely on various analytical methods to confirm particle size, shape, identity markers (CD63, CD9, CD81), absence of non-vesicular contaminants, sterility, mycoplasma testing, endotoxin levels, residual DNA content, and, when needed, the amount of engineered or foreign materials [247]. Stability poses another challenge since EVs are sensitive to freeze-thaw cycles and long-term storage. Freezing EVs at −80 °C increases their size, but freeze-thaw cycles reduce their population over time [251]. Lyophilization with cryoprotectants, controlled-temperature logistics, and optimized buffer formulations are increasingly used to maintain vesicle function during storage and transport [252]. In this context, lung-derived EVs loaded with severe acute respiratory syndrome coronavirus 2 (SARS-CoV-2) spike protein-encoding mRNA were lyophilized to extend the shelf life. The formulation has remained stable at room temperature and has shown superior therapeutic efficiency compared with liposome-based formulations [253].

From a production perspective, scalability depends on using controlled bioreactor systems such as hollow-fiber bioreactors or microcarrier suspension cultures along with downstream purification methods like tangential-flow filtration, size-exclusion chromatography, or affinity capture [254]. Automation and closed-system manufacturing are being integrated gradually to reduce batch-to-batch variability and enhance GMP compliance.

Regulatory frameworks for EV therapeutics are continually changing. EVs from minimally manipulated primary cells are usually classified as biological products, while engineered EVs that carry therapeutic nucleic acids or surface ligands might be regulated as advanced therapy medicinal products (ATMPs). These require more detailed documentation and testing. Regulatory agencies need complete descriptions of the manufacturing cell line’s origin and history, along with control of critical quality attributes (CQAs), validated potency assays, and preclinical biodistribution and pharmacokinetic studies. These studies assess organ accumulation, clearance, and off-target risks [255]. Labeling of EVs could be a suitable technique for assessing the displacement of EVs into the body. However, the use of labelling probes can modify the features of EVs and consequently should be considered as well as the sophisticated and expensive resources for individualize their tracking [256]. Risk assessments must cover viral safety, leftover vector components, horizontal gene transfer, and the genomic stability of engineered parent cells. The lack of standardized EV tracking methods makes biodistribution assessments tricky, leading to a need for unified imaging and labeling strategies.

Engineering strategies strongly impact the effectiveness of EV-based therapeutics. Genetic engineering of parent cells can improve targeting, increase therapeutic cargo loading, or alter immunological properties. However, this requires thorough proof of genomic stability and ensuring there are no replication-competent vector contaminants. Chemical modifications to surfaces can allow for targeted delivery, but these changes may cause immunogenic epitopes or complicate purification and lower manufacturing yields [250]. Physical loading strategies like electroporation, extrusion, or sonication can enhance cargo encapsulation but may risk damaging membrane integrity or creating aggregates that need to be removed to meet purity standards [257]. Hybrid EVs or biomimetic vesicles that use synthetic materials add more regulatory uncertainty regarding biodegradation, biocompatibility, and classification paths. Therefore, engineering decisions must find a balance between therapeutic benefits and manufacturability, safety, scalability, and overall regulatory likelihood.

It is important to note that several years have passed since the emergence of the approach to using EVs as DDSs. However, at present, no EV-based formulations have been approved by government health agencies. This finding indicates a requirement for additional observation to determine the feasibility of implementing these formulations in clinical practice. Notwithstanding the encouraging applications, the brief duration of action, comprehensive understanding of their genesis, control of drug encapsulation, and specific targeting are the principal challenges to be addressed are the therapeutic activities of these agents, which are associated with anticancer drugs and gene or epigenetic therapies. The manufacturing of a safe EV necessitates the implementation of control stages during the isolation and purification phases, as well as post-modification (engineering). Furthermore, consensus must be reached on the doses administered, taking into consideration the type of EV, the treatment modality, and the disease state. Consequently, it is hypothesized that the progression of knowledge in this domain is accelerating, and in the near future, products based on EVs will be near arrival [258,259].

## 10. Challenges and Future Perspectives

Despite recent advances, the clinical application of extracellular vesicles (EVs) still faces significant challenges, including low large-scale production, lack of standardization in isolation and characterization methods, heterogeneity of EV populations, low drug-loading efficiency, and concerns regarding toxicity or off-target effects of modifications. For example, decorating EVs with RGD peptides to enhance uptake by tumor cells (cells that express integrins) while reducing interaction with healthy tissues. To reduce immunogenic reactions, choosing the right parental cell type minimizes risks of immune activation or inflammatory responses, for example, mesenchymal cells. The seek for a correct strategy for loading drugs or genetic material can lead to leaky membrane vesicles or unstable, which could release their material in a burst mode, with consequences for patients because of the significant activity of siRNA or chemotherapeutic agents. For this reason, gentle loading mechanisms should be implemented to prevent the generation of an unstable membrane. The future of EVs as drug delivery vehicles lies in the integration of native and engineered approaches to harness their inherent advantages while overcoming current limitations. Overcoming the heterogeneity and purity problems is an important challenge that implies improving current isolation techniques often concomitantly entails contaminants like proteins or lipoproteins. Lack of purity impacts the reduction of reproducibility and complicates regulatory approval for therapeutic use [40,260].

Finally, the clinical translation of EVs poses a challenge due to the numerous regulatory hurdles they must overcome. The establishment of strict frameworks to be applied in therapeutics implies the development of approval protocols for designing, production, testing, and usage form, to guarantee the safety and effectiveness for patients. One of the first steps could be to reach a homogeneous classification, considering their physicochemical properties (lipid bilayer vesicles), but also their biological origin. This last point is also crucial because it ranges from type producer cell (for example, adipose mesenchymal cell) and cancerous or healthy. Therefore, novel categories in drug delivery systems could be defined or addressing unique features and functions.

## 11. Conclusions

EVs are preferred when the therapeutic goal requires sophisticated biological interactions—such as crossing the blood–brain barrier, avoiding immune detection, or leveraging innate homing abilities—that are incredibly difficult to engineer from scratch into synthetic systems.

They represent a shift from “bottom-up” engineering to a “biology-first” approach, harnessing nature’s own optimized delivery mechanisms. While synthetic carriers like LNPs (as used in COVID-19 mRNA vaccines) excel in scalable, reproducible production for widespread use, EVs hold immense promise for delivering the next generation of precision medicines, especially for complex diseases like cancer, neurological disorders, and rare genetic conditions.

In the end, moving EV-based nanodevices into clinical use will depend on combining clinical insights, strong CMC workflows, unified regulatory expectations, and engineering strategies that work well in real-world manufacturing. As clinical evidence grows and good manufacturing practices evolve, EVs are set to become a new generation of versatile, biologically informed nanotherapeutics. However, their successful transition will need to tackle the linked challenges of standardization, defining potency, producing at a large scale, and ensuring safety.

## Figures and Tables

**Figure 1 pharmaceutics-17-01617-f001:**
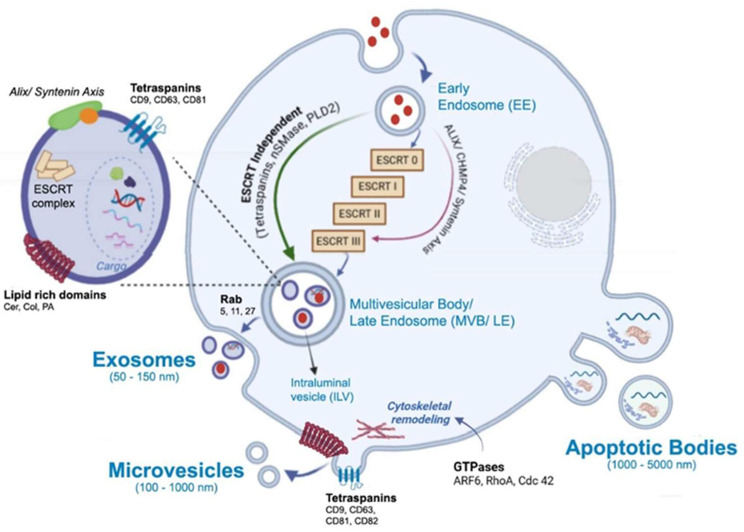
Biogenesis of extracellular vesicles and main components which comprises membrane and cytosolic proteins (tetraspanins, enzymes, receptors), lipids (ceramide, cholesterol, phosphatidylserine), metabolites and amino acids, and nucleic acids, DNA and a range of RNA species: messenger RNAs (mRNAs), microRNAs (miRNAs), long non-coding RNAs (lncRNAs), circular RNAs (circRNAs). The schematic illustrations were created with BioRender.com. Benedini, L. (2025) https://BioRender.com/gy6255j.

**Figure 2 pharmaceutics-17-01617-f002:**
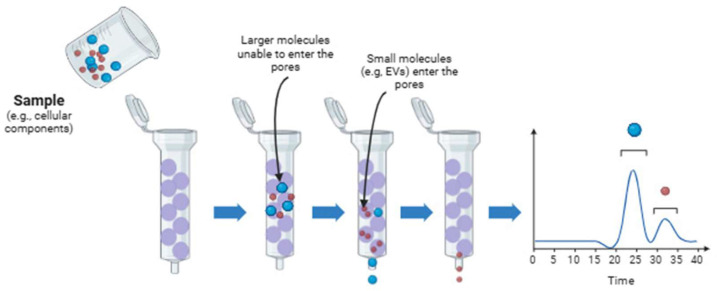
A schematic representation of the size exclusion chromatography technique. The schematic illustrations were created with BioRender.com. Benedini, L. (2025) https://BioRender.com/079h6bh.

**Figure 3 pharmaceutics-17-01617-f003:**
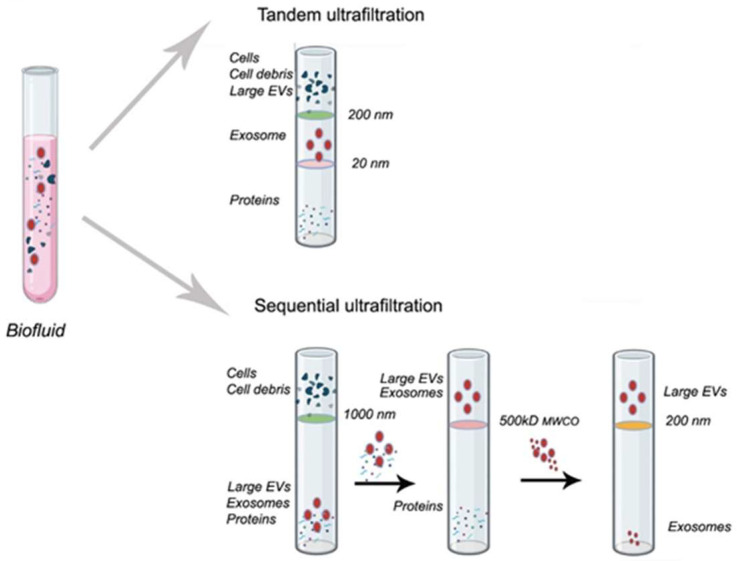
A schematic representation of the filtration technique. The schematic illustrations were created with BioRender.com. Benedini, L. (2025) https://BioRender.com/59laya3.

**Figure 4 pharmaceutics-17-01617-f004:**
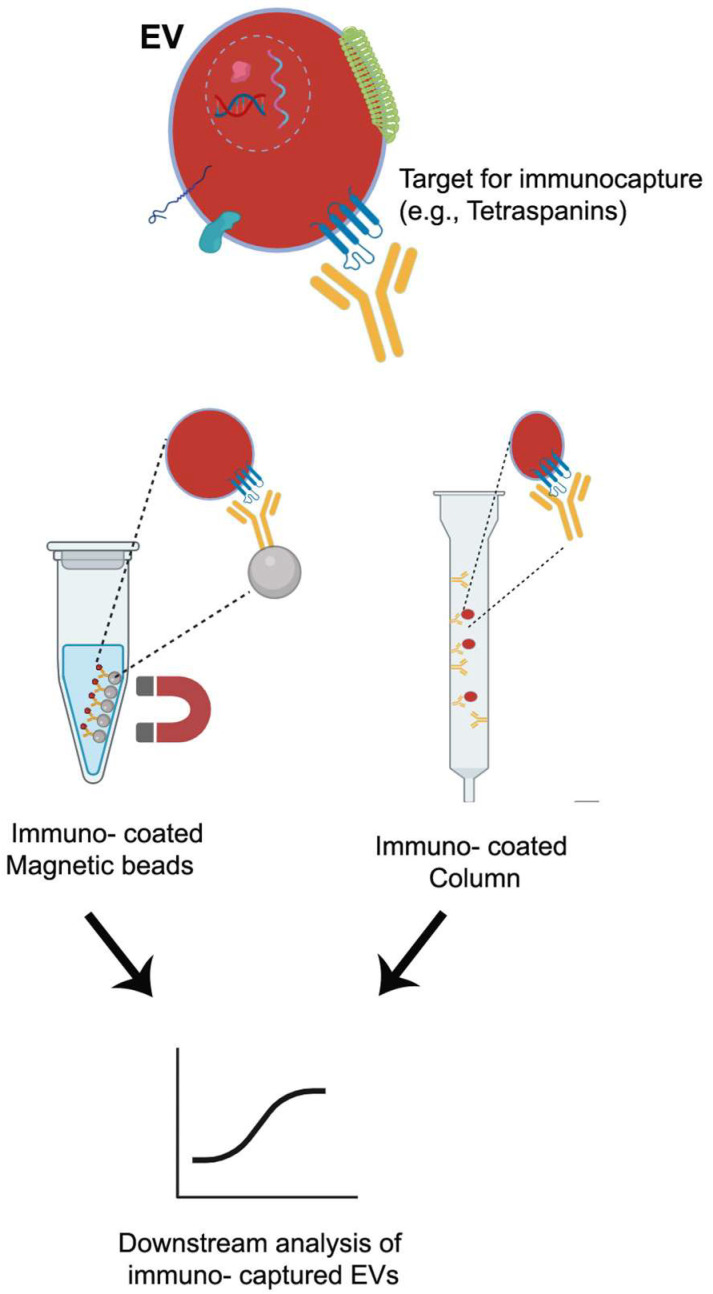
Affinity-based techniques. The schematic illustrations were created with BioRender.com. Benedini, L. (2025) https://BioRender.com/f6srf1l.

**Figure 5 pharmaceutics-17-01617-f005:**
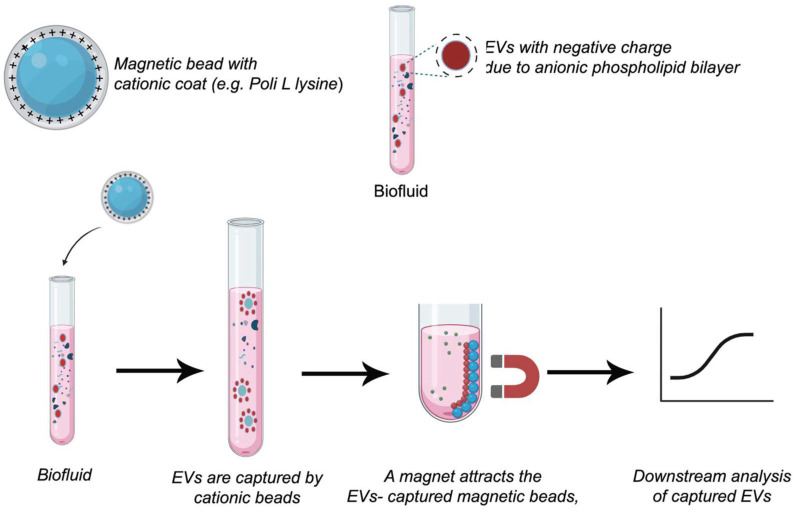
Ion-exchange techniques. The schematic illustrations were created with BioRender.com. Benedini, L. (2025) https://BioRender.com/jqreh03.

**Figure 6 pharmaceutics-17-01617-f006:**
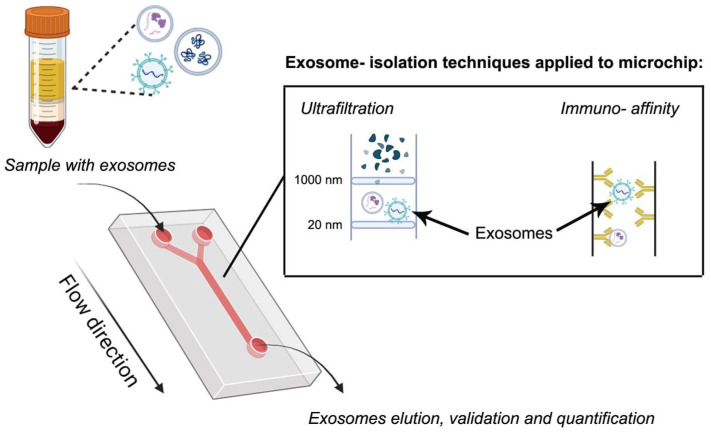
Microfluidic technique based on ultrafiltration and immunoaffinity. The schematic illustrations were created with BioRender.com. Benedini, L. (2025) https://BioRender.com/zbe5tb5.

**Table 1 pharmaceutics-17-01617-t001:** Molecular sorting and packaging mechanisms of nucleic acids into extracellular vesicles (EVs), including exosomes (EXs) and microvesicles (MVs).

Nucleic Acid Type	Subtype/Features	Origin	Sorting Mechanism	Key Binding Proteins	Functional Implications	References
RNA	miRNAs (e.g., GGAG motif)	Nuclear/Cytoplasmic	Sequence-specific recognition and PTM-dependent sorting	hnRNPA2B1 (SUMOylated, O-GlcNAcylated)	Gene regulation in recipient cells	[30,73]
	tRNA fragments, Y RNAs, long mRNAs	Cytoplasmic	Phase-separated condensates targeting MVBs	YBX1 (PTM-sensitive)	Stress response, RNA stability	[31,74]
	miRNAs (RISC-bound)	Cytoplasmic	Signal-dependent phosphorylation and localization	AGO2 (KRAS–MEK–ERK modulated)	Silencing, oncogenic signaling	[75,76]
	snoRNAs, miRNAs	Nuclear/Cytoplasmic	Purine-rich motif recognition, endosomal targeting	hnRNPK (interacts with caveolin-1)	Modulation of recipient cell transcriptome	[77]
	General RNA subsets	Plasma membrane microdomains	Lipid raft and actin scaffold association	hnRNPA2B1, YBX1 (in microvesicles)	Functional heterogeneity of EVs	[78,79]
DNA	Genomic DNA (dsDNA, ssDNA, chromatin fragments)	Nuclear	Micronucleus rupture, autophagy-linked capture	Histones, HMGB1/2	Immunostimulation, oncogenic transfer	[80]
	Mitochondrial DNA (mtDNA)	Mitochondria	Nucleoid protein-mediated stabilization and targeting	TFAM	Mitochondrial signaling, immune activation	[81]
	DNA–protein complexes	Cytosolic/Endosomal	ESCRT-associated recruitment (TSG101, ALIX)	TSG101, ALIX	Vesicular DNA integrity and packaging	[80]
	Stress-induced DNA	Nuclear/Cytosolic	γH2AX-marked damage response	γH2AX, candidate nucleases (exploratory)	cGAS–STING activation, inflammation	[82]

## Data Availability

The raw data supporting the conclusions of this article will be made available by the authors on request.

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
