# Peer review of "Redefining the Limits of Nanodevices-Based Drug Delivery Systems: Extracellular Vesicles"

_pharmaceutics, 2025, doi:10.3390/pharmaceutics17121617_

Round 1

Reviewer 1 Report

Comments and Suggestions for Authors

This manuscript presents a timely and comprehensive review on extracellular vesicles and attempts to frame them as natural nanodevices for drug delivery. That said, the current version still reads more like a biologically oriented EV overview than a pharmaceutics-oriented, problem-driven review on EV-based drug delivery systems. The following issues need to be addressed before acceptance:

  1. The introduction provides a comprehensive overview of the biological background of extracellular vesicles; however, it lacks a focused discussion on their relevance to drug delivery systems, which could be strengthened.
  2. The transitions between major sections are not always smooth. The addition of transitional sentences would help connect topics logically and summarize key takeaways.
  3. The biological aspects of EVs are described in great detail, yet their pharmaceutical implications and relevance to drug delivery performance are not sufficiently emphasized.
  4. Although the classification of EVs is well described, the differences in drug loading strategies among various EV subtypes are not clearly presented.
  5. The review’s central theme, conceptualizing EVs as natural nanodevices, is intriguing. However, the theoretical foundation and true innovative aspects of this concept should be discussed in greater depth.
  6. Figures 1–6 primarily serve as schematic illustrations. Their sources or self-drawing declarations should be clearly indicated.
  7. In the “Therapeutic Approaches” section, the categorization of drug types and corresponding loading methods should be expanded.
  8. The discussion on clinical applications and challenges of EV-based therapeutics is relatively brief. Supplementing this part with more detailed.
  9. The future perspectives section is somewhat general. It would benefit from more specific directions and emerging research trends.
  10. Excessive self-citation from the authors’ own previous work may affect the objectivity of the review. It is recommended to balance citations by including more recent and representative publications from the broader field.
  11. Technical terminology should be standardized throughout the manuscript (e.g., consistent use of “exosomes”“EXs,” and “drug delivery system” vs. “nanocarrier”).
  12. The English writing is generally fluent, but there are repetitive sentence structures and minor grammatical issues. A thorough language polishing would further improve readability and professionalism
Comments on the Quality of English Language

The manuscript is generally written in fluent and understandable English; however, there are several minor grammatical errors and repetitive sentence structures throughout the text. Certain long sentences could be shortened for clarity, and technical terms should be used consistently. A careful language editing and proofreading by a native or professional scientific editor is recommended to improve readability and precision.

Author Response

First, I want to thank you for taking the time to review the manuscript and for your attentive comments. 

Comments and questions were answered one by one here. Clarifying texts were included in the main manuscript in blue.

Comments and Suggestions for Authors

This manuscript presents a timely and comprehensive review of extracellular vesicles and attempts to frame them as natural nanodevices for drug delivery. That said, the current version still reads more like a biologically oriented EV overview than a pharmaceutics-oriented, problem-driven review on EV-based drug delivery systems. The following issues need to be addressed before acceptance:

  1. The introduction provides a comprehensive overview of the biological background of extracellular vesicles; however, it lacks a focused discussion on their relevance to drug delivery systems, which could be strengthened.

R: The discussion of the relevance of EVs as DDSs has been deepened and improved.

  1. The transitions between major sections are not always smooth. The addition of transitional sentences would help connect topics logically and summarize key takeaways.

R: Transitional paragraphs were included to make the reading smooth and fluent.

  1. The biological aspects of EVs are described in great detail, yet their pharmaceutical implications and relevance to drug delivery performance are not sufficiently emphasized

R: Section 5, "Pharmaceutical implications of the biological features of EVs", was included in the text. This section describes the relation between the biological structures of EVs and their relevance in drug delivery systems.

  1. Although the classification of EVs is well described, the differences in drug loading strategies among various EV subtypes are not clearly presented.

R: You are right. The differences in drug loading strategies among various EV subtypes are not clearly presented. For this reason, they were improved. We have included section 8.2.1. Loading methods of active agents.

  1. The review’s central theme, conceptualizing EVs as natural nanodevices, is intriguing. However, the theoretical foundation and true innovative aspects of this concept should be discussed in greater depth.

R: The theoretical foundation and true innovative aspects of EVs as DDSs have been deeply discussed, as the reviewer suggests, mainly in the introduction and in sections 7 and 8.

  1. Figures 1–6 primarily serve as schematic illustrations. Their sources or self-drawing declarations should be clearly indicated.

R: You are right. The figures have been carried out for the authors. An explanatory text has been included in all schematic figures.

  1. In the “Therapeutic Approaches” section, the categorization of drug types and corresponding loading methods should be expanded.

R: Categorization and loading methods for different therapeutic agents were included such as the reviewer suggests.

  1. The discussion on clinical applications and challenges of EV-based therapeutics is relatively brief. Supplementing this part with more detailed.

R: The clinical application and challenges of EVs were supplemented as the reviewer suggests. Section 9. Clinical translation was included in the main text.

  1. The future perspectives section is somewhat general. It would benefit from more specific directions and emerging research trends.

R: Future perspectives section was improved and focused on emerging trends.

  1. Excessive self-citation from the authors’ own previous work may affect the objectivity of the review. It is recommended to balance citations by including more recent and representative publications from the broader field.

R: All the references have been revised. Among more than 200 references, we have included only 2 self-citations Benedini, L. & Messina, P. (2022). Lipid-based nanocarriers for drug delivery: microemulsions versus nanoemulsions. In Systems of Nanovesicular Drug Delivery (pp. 39–53). Elsevier. https://doi.org/10.1016/B978-0-323-91864-0.00001-2 and D´elĺa, N, Gravina, A. N., Benedini, L. & Messina, P. V. (2024). A commentary: harnessing vesicles power with new scenes of membrane-based devices for drug delivery. Biocell, 48(10), 1401–1403. https://doi.org/https://doi.org/10.32604/biocell.2024.055512.

  1. Technical terminology should be standardized throughout the manuscript (e.g., consistent use of “exosomes”“EXs,” and “drug delivery system” vs. “nanocarrier”).

R: This is an accurate commentary. The terminology has been standardized in the manuscript.

  1. The English writing is generally fluent, but there are repetitive sentence structures and minor grammatical issues. A thorough language polishing would further improve readability and professionalism.

R: The English writing was revised and improved.

Reviewer 2 Report

Comments and Suggestions for Authors

Specific comments are as follows:

1. Many parts of the manuscript’s statements or discussion lack citation support. For example, the authors claim: “Synthetic nanocarriers, most notably liposomes and polymeric nanoparticles, have demonstrated considerable progress in enhancing drug pharmacokinetics. However, these carriers frequently encounter challenges related to immunogenicity, inefficient cellular uptake, rapid clearing by the mononuclear phagocyte system, and an absence of particular tissue targeting.” Please provide supporting references for each clause, and audit the entire manuscript to ensure all nontrivial assertions and quantitative details are appropriately and currently referenced.

2. Section 5 (Isolation and Purification Methods) largely reiterates previous reviews; please clarify the novel insights offered here.

3. Sections 6 and 7 appear overlapped and confusing; please streamline to avoid duplication.

4. Structure of section 7 was poorly arranged. The discussion jumps into MSC EVs and their engineering while other cell sources are not discussed. Either broaden the scope (e.g., dendritic, macrophage, epithelial, tumor-derived, milk, bacterial EVs) and discuss whether they are therapeutic candidates and how engineering strategies translate to them, or state upfront that MSC EVs are the focus and explain why. References are missing throughout, especially for large-cargo loading where practical constraints matter; please cite methods and performance metrics and add examples for mRNA (e.g., PMID: 37847907) and Cas9/large protein cargo (e.g., PMID: 38486996).

5. Clinical translation is insufficiently discussed. Please expand on the current clinical landscape of EVs as drug-delivery systems (trials, phases, indications, routes, dosing, endpoints, safety), key CMC/GMP considerations (potency assays, purity/release criteria, stability), regulatory expectations, and how engineering choices impact manufacturability and risk, particularly for engineering therapeutic extracellular vesicles for clinical translation.

Author Response

First, I want to thank you for taking the time to review the manuscript and for your attentive comments

Comments and questions were answered one by one here. Clarifying texts were included in the main manuscript in blue.

Comments and Suggestions for Authors

Specific comments are as follows:

  1. Many parts of the manuscript’s statements or discussion lack citation support. For example, the authors claim: “Synthetic nanocarriers, most notably liposomes and polymeric nanoparticles, have demonstrated considerable progress in enhancing drug pharmacokinetics. However, these carriers frequently encounter challenges related to immunogenicity, inefficient cellular uptake, rapid clearing by the mononuclear phagocyte system, and an absence of particular tissue targeting.” Please provide supporting references for each clause, and audit the entire manuscript to ensure all nontrivial assertions and quantitative details are appropriately and currently referenced.

R: All references have been revised. In the previous paragraph has been included the references show below. “Synthetic nanocarriers, most notably liposomes and polymeric nanoparticles, have demonstrated considerable progress in enhancing drug pharmacokinetics (D. Chen et al., 2023). However, these carriers frequently encounter challenges related to immunogenicity, inefficient cellular uptake, rapid clearing by the mononuclear phagocyte system, and an absence of particular tissue targeting (Karmaker et al., 2025)”.

  1. Section 5 (Isolation and Purification Methods) largely reiterates previous reviews; please clarify the novel insights offered here.

R: Novel insights about this topic were included, as the reviewer suggests.

  1. Sections 6 and 7 appear overlapped and confusing; please streamline to avoid duplication.

R: You are right. The sections were renumbered and improved.

  1. Structure of section 7 was poorly arranged. The discussion jumps into MSC EVs and their engineering while other cell sources are not discussed. Either broaden the scope (e.g., dendritic, macrophage, epithelial, tumor-derived, milk, bacterial EVs) and discuss whether they are therapeutic candidates and how engineering strategies translate to them, or state upfront that MSC EVs are the focus and explain why. References are missing throughout, especially for large-cargo loading where practical constraints matter; please cite methods and performance metrics and add examples for mRNA (e.g., PMID: 37847907) and Cas9/large protein cargo (e.g., PMID: 38486996).

R: It is a precise suggestion. The structure of the mentioned section was improved by the addition of a deep discussion of EVs from different origins, and the references were accurately cited.

  1. Clinical translation is insufficiently discussed. Please expand on the current clinical landscape of EVs as drug-delivery systems (trials, phases, indications, routes, dosing, endpoints, safety), key CMC/GMP considerations (potency assays, purity/release criteria, stability), regulatory expectations, and how engineering choices impact manufacturability and risk, particularly for engineering therapeutic extracellular vesicles for clinical translation.

R: You are right. This topic is practically without discussion in the last section. The clinical application and challenges of EVs were supplemented, as the reviewer suggests. Section 9. Clinical translation was included in the main text.

Round 2

Reviewer 1 Report

Comments and Suggestions for Authors

The authors have thoroughly addressed most of the previous comments, with substantial improvements in the sections on drug delivery mechanisms, engineering strategies, GMP considerations, large-scale production, and regulatory challenges. These revisions have made the review more complete and substantially stronger. I believe the manuscript, in its current form, is suitable for acceptance.
However, one further refinement would be beneficial:
Given the pharmaceutical focus of Pharmaceutics, I recommend that the authors make a modest adjustment to the introduction to strengthen its central narrative around drug delivery, ensuring that the review’s main theme is clearly aligned with the journal’s scope.

Author Response

Reviewer 1

Comments and Suggestions for Authors

The authors have thoroughly addressed most of the previous comments, with substantial improvements in the sections on drug delivery mechanisms, engineering strategies, GMP considerations, large-scale production, and regulatory challenges. These revisions have made the review more complete and substantially stronger. I believe the manuscript, in its current form, is suitable for acceptance. However, one further refinement would be beneficial: Given the pharmaceutical focus of Pharmaceutics, I recommend that the authors make a modest adjustment to the introduction to strengthen its central narrative around drug delivery, ensuring that the review’s main theme is clearly aligned with the journal’s scope.

R: We have modified the manuscript to focus on the importance of extracellular vesicles in the pharmaceutical context. A text is included in the Introduction section.

Reviewer 2 Report

Comments and Suggestions for Authors

The authors have satisfactorily addressed most of my concerns. Only one minor suggestion on the conlcusion section. The discussion of trade-offs associated with EV engineering (EV loading) can be expanded to address scalability, complexity, potential contamination introduced during the loading process, and overall cost-effectiveness, and that recent work on bioengineered EVs (EV loading), particularly studies addressing clinical translation (PMID: 39227240, PMID: 37421185), can be be introduced and discussed as well.

Author Response

Reviewer 2

Comments and Suggestions for Authors

The authors have satisfactorily addressed most of my concerns. Only one minor suggestion on the conlcusion section. The discussion of trade-offs associated with EV engineering (EV loading) can be expanded to address scalability, complexity, potential contamination introduced during the loading process, and overall cost-effectiveness, and that recent work on bioengineered EVs (EV loading), particularly studies addressing clinical translation (PMID: 39227240, PMID: 37421185), can be be introduced and discussed as well.

R: We have improved the discussion of trade-offs associated with EV engineering. Texts are added in this concern in Section 9, clinical translation of EVs, and the suggested references are included
